# Orthogonal replication with optogenetic selection evolves yeast *JEN1* into a mevalonate transporter

Scott A Wegner[1], Virginia Jiang [ID][2], Jeremy D Cortez[1] & José L Avalos [ID][1,2,3,4,5]✉

## Abstract

**The in vivo continuous evolution system OrthoRep (orthogonal replication) is a powerful strategy for rapid enzyme evolution in *Saccharomyces cerevisiae* that diversifies genes at a rate exceeding the endogenous genome mutagenesis rate by several orders of magnitude. However, it is difficult to neofunctionalize genes using OrthoRep partly because of the way selection pressures are applied. Here we combine OrthoRep with optogenetics in a selection strategy we call OptoRep, which allows fine-tuning of selection pressure with light. With this capability, we evolved a truncated form of the endogenous monocarboxylate transporter *JEN1 (JEN1t)* into a de novo mevalonate importer. We demonstrate the functionality of the evolved *JEN1t (JEN1t^{Y180C/G})* in the production of farnesene, a renewable aviation biofuel, from mevalonate fed to fermentation media or produced by microbial consortia. This study shows that the light-induced complementation of OptoRep may improve the ability to evolve functions not currently accessible for selection, while its fine tunability of selection pressure may allow the continuous evolution of genes whose desired function has a restrictive range between providing effective selection and cellular viability.**

**Keywords** OrthoRep; Optogenetics; Continuous Evolution; Mevalonate Transporter Evolution; *JEN1*
**Subject Categories** Evolution & Ecology; Metabolism; Microbiology, Virology & Host Pathogen Interaction

## Introduction

OrthoRep is a valuable tool that combines the advantages of both adaptive and directed laboratory evolution. Traditional adaptive laboratory evolution allows for the unbiased evolution of function through the application of appropriate selection pressure (Dragosits and Mattanovich, 2013; Fernandes et al, 2023). However, these evolution campaigns can be lengthy due to the high fidelity of endogenous DNA replication, which has error rates of $10^{-10}$–$10^{-9}$ substitutions per base pair (Lang and Murray, 2008; Ravikumar

et al, 2018). To address this, directed evolution often involves in vitro mutagenesis of a gene target, resulting in higher mutation frequencies but requiring multiple rounds of diversification and enrichment (Cobb et al, 2013). Consequently, classic protein evolution methods are limited either in depth (adaptive laboratory evolution) or in scale (directed evolution). OrthoRep overcomes these limitations by applying in vivo hypermutation in *Saccharomyces cerevisiae* to a selected gene target at an error rate orders of magnitude higher than that of the host genome (Molina et al, 2022).

To hypermutagenize a target gene with OrthoRep, the system utilizes the p1/p2 killer plasmid pair from *Kluyveromyces lactis*, stably maintained in *S. cerevisiae*. The target gene of interest (GOI) is inserted in the p1 plasmid, while the p2 plasmid contains the RNA polymerase to achieve cytosolic transcription of the GOI and a high-fidelity DNA polymerase (TP-DNAP2) to replicate p2. In addition, p1 is replicated by a specialized cytosolic DNA polymerase (TPDNAP1) heterologously expressed from a nuclear plasmid (Kämper et al, 1989). By utilizing an engineered error-prone TPDNAP1-4-2 DNA polymerase, a focused mutagenesis rate of $1 \times 10^{-5}$ substitutions per base pair (s.p.b.) is achieved in the p1-encoded GOI (Ravikumar et al, 2018). This error rate enables targeted mutagenesis at levels higher than those causing genomic extinction while preserving the host genome's natural fidelity (Ravikumar et al, 2018). The OrthoRep system can be transferred to commonly used yeast backgrounds to host the hypermutation of GOIs and direct their evolution (Javanpour and Liu, 2019). The ease and flexibility of transferring its components, and the high error rate that spares the host genome, make OrthoRep a powerful tool for continuous in vivo evolution in yeast.

Despite these advantages, OrthoRep generally uses activity-dependent growth selection to enrich functional mutants, which can introduce some limitations. This method typically requires genetic complementation, which involves replacing an essential metabolic function with a gene of interest that exhibits some basal level of that functionality. Moreover, it requires the ability to supplement an essential metabolite downstream from the GOI to enable growth and p1 mutagenesis, which constrains the genes that can be evolved. For example, OrthoRep was used to evolve TrpB, an enzyme that couples L-serine and indole to produce L-tryptophan, to function without its heterodimeric allosteric activator TrpA (Rix et al, 2020, 2024). Mutagenesis of TrpB was directed to complement a *TRP5* deletion by stepwise decreases in L-tryptophan

[1]Department of Molecular Biology, Princeton University, Princeton, NJ 08544, USA. [2]Department of Chemical and Biological Engineering, Princeton University, Princeton, NJ 08544, USA. [3]The Omenn-Darling Bioengineering Institute, Princeton University, Princeton, NJ 08544, USA. [4]The Andlinger Center for Energy and the Environment, Princeton University, Princeton, NJ 08544, USA. [5]High Meadows Environmental Institute, Princeton University, Princeton, NJ 08544, USA. ✉E-mail: javalos@princeton.edu

supplementation, resulting in new TrpB variants with higher overall activity and promiscuous activity for new indole derivatives (Rix et al, 2020). This was possible because yeast naturally imports L-tryptophan, meaning TrpB did not need to be active initially to overcome the *TRP5* auxotrophy. Although methods have been developed for OrthoRep that do not require activity-dependent growth selection, such as enrichment through fluorescent sorting, not all GOIs can be evolved with these methods (Jensen et al, 2021; Wellner et al, 2021). Moreover, while it has been suggested that new variants of OrthoRep can evolve de novo activity in GOIs, general OrthoRep strategies still necessitate constraining yeast growth to an existing basal level of desired enzymatic activity (Rix et al, 2024). Thus, achieving gain-of-function mutations that neofunctionalize genes (confer a completely new activity) with OrthoRep remains challenging due to the general requirement of basal activity in the GOI to complement an essential function in yeast.

Engineering mevalonate import activity in *S. cerevisiae* requires gain-of-function mutations that neofunctionalize existing transporters because none have been identified with even basal levels of such activity. Mevalonate is a key metabolite, not only as a precursor of ergosterol (a fungal equivalent to cholesterol), but also as a precursor in the biosynthesis of isoprenoids, which are the largest family of natural products, including fragrances, flavors, pigments, bioactive compounds of pharmaceutical importance, and even biofuels (Sacchettini and Poulter, 1997; Vickers et al, 2017). Despite its importance and the existence of mevalonate import systems in several bacterial species, there is no evidence for mevalonate import, and no mevalonate importer has been identified in *S. cerevisiae* (Rodriguez et al, 2016). This complicates the traditional complementation approach used in OrthoRep given that mevalonate feeding and selection pressure by constraining this feeding (decremental mevalonate media concentrations) are not viable options. In addition, the only previously known mevalonate importer is a mutant of the mammalian monocarboxylate transporter MCT1 (F360C). However, its activity in yeast is very low due to poor membrane trafficking, making it inadequate to evolve mevalonate-dependent growth in *S. cerevisiae* (Makuc et al, 2004; Wegner et al, 2024). Therefore, a better option to evolve a mevalonate importer in yeast could be to mutagenize an endogenous monocarboxylate transporter, such as *JEN1*, to avoid having to select for both improved trafficking as well as mevalonate importation. This, however, poses two challenges to OrthoRep: (1) requiring neofunctionalization through gain-of-function mutations; and (2) directing the continuous evolution process without a basal importation activity that would allow using mevalonate feeding as a strategy to exert increasing selective pressure.

In this study, we combined OrthoRep with optogenetics to tackle these challenges in a strategy we call OptoRep and used it to evolve *JEN1* into a mevalonate transporter. To achieve this, we used the light-responsive EL222 transcription factor from *Erythrobacter litoralis*, which contains a blue light-activated (450 nm) LOV (Light Oxygen Voltage) domain, to control endogenous mevalonate biosynthesis, thereby fine-tuning the selection pressure applied to OrthoRep by light-induced gene expression. OptoRep was able to select for gain-of-function mutations that neofunctionalize *JEN1* into an effective mevalonate importer. This acquired function is robust enough to improve the production of farnesene (a renewable aviation fuel) by feeding mevalonate to engineered strains or in microbial consortia with mevalonate-secreting strains. OptoRep can thus facilitate the evolution of gain-of-function mutations that neofunctionalize genes through OrthoRep-enabled continuous evolution, using light to fine-tune the selection pressure.

# Results

## Construction of a light-sensitive mevalonate auxotrophic strain (OptoMEV)

To fine-tune the sensitivity of yeast to the selection pressure caused by mevalonate insufficiency, we developed a light-sensitive mevalonate auxotrophic strain, whose ability to synthesize mevalonate depends on light. To achieve this, we used the OptoEXP optogenetic system, which uses the blue light-activated transcription factor EL222 from *Erythrobacter litoralis* (Fig. 1A,B), to control the expression of *HMG1* (HMG-coa reductase), the enzyme that produces mevalonate (Zhao et al, 2018). We replaced the native *HMG1* promoter with the cognate EL222 promoter ($P_{C120}$) using CRISPR-Cas9, and deleted *HMG2*, the *HMG1* paralog (Basson et al, 1986). Deletion of *HMG2* is necessary as it alone provides sufficient HMG-CoA reductase activity for growth (Basson et al, 1986). The resulting strain, termed OptoMEV for optogenetic conditional mevalonate auxotroph, grows under permissive blue light conditions at a similar rate as the wild-type and single $\Delta hmg2$ strains (Fig. 1C). Under blue light conditions, we measured growth rates of $0.286 \pm 0.033\,h^{-1}$ and $0.251 \pm 0.024\,h^{-1}$ for the wild-type and OptoMEV strains respectively (Appendix Table S1). However, in the dark, OptoMEV shows a severe growth defect, unlike the wild-type and $\Delta hmg2$ strains (Fig. 1D; Appendix Table S1), achieving only ~8% of its growth rate in blue light. The low starting inocula (0.01 $OD_{600}$) helped remove residual Hmg1p from overnight cultures through mitotic dilution. These results are consistent with OptoMEV being unable to grow in the dark because under these conditions it becomes deficient in mevalonate, required for ergosterol biosynthesis and thus growth.

We next tested whether the growth of OptoMEV in the dark could be rescued by feeding mevalonate to the growth media. When we added 10 mM mevalonate to the growth medium, we observed a negligible effect on the growth rate of OptoMEV in the dark (Fig. 1D,F; Appendix Table S1). Mevalonate feeding had no effect on the growth of OptoMEV in blue light or the growth of control strains in either light condition. To verify that native *JEN1* is not involved in mevalonate importation, we tested the $\Delta jen1$ OptoMEV strain for growth and found it to be practically indistinguishable from its parent OptoMEV, whether mevalonate is fed to the media or not, and regardless of light conditions (Fig. 1C–F; Appendix Table S1). Furthermore, even with mevalonate in the media, OptoMEV still grows about 10 times slower than the wild type, with residual growth in dark conditions likely due to leaky *HMG1* expression from $P_{C120}$, which has been previously reported (Zhao et al, 2018), but is still too low to enable the use of OrthoRep to evolve a mevalonate importer.

## OptoRep yields a mevalonate importer through a gain-of-function mutation in *JEN1*

Because of its natural monocarboxylate transporter activity, we reasoned that *JEN1* was a good candidate from which to evolve a

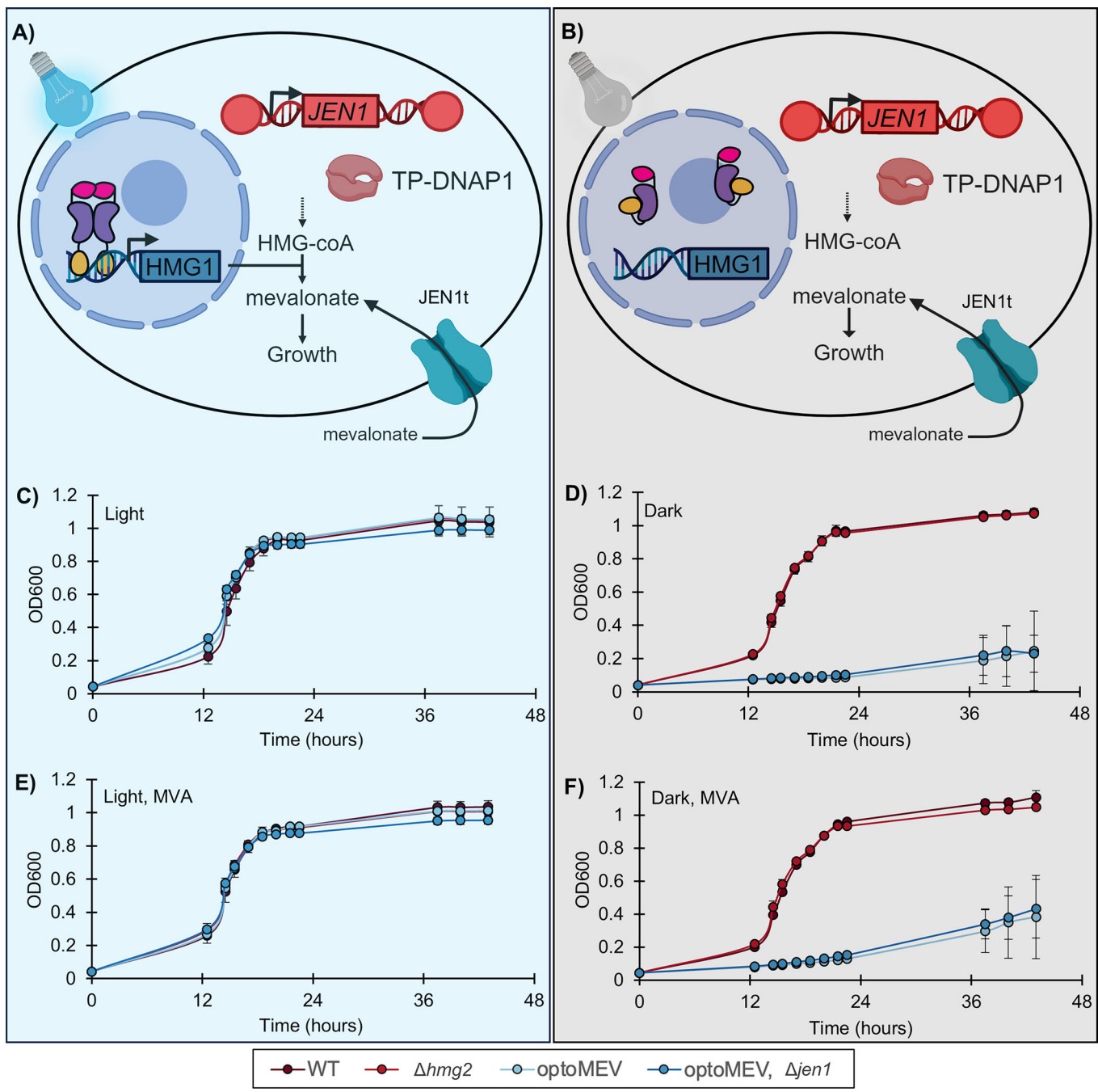

**Figure 1. Generation of the OptoMEV strain.**

(A, B) Optogenetic control of *HMG1* using a blue light-sensitive EL222-VP16 fusion transcription factor regulates growth. Under blue light, EL222 dimerizes, which allows it to bind to $P_{C120}$ promoter and activate *HMG1* transcription to allow growth. In darkness, OptoMEV can only grow by importing mevalonate from the media, setting a selection strategy to evolve *JEN1t* into a mevalonate importer by tuning the selection pressure with light. Growth curves of the wild-type-like parent strain (WT; SAWy119), a strain in which the paralog of *HMG1* is deleted (*Δhmg2*; SAWy518), the OptoMEV strain ($P_{C120}$::*HMG1*, *Δhmg2*; SAWy524), and the OptoMEV strain with a *JEN1* deletion ($P_{C120}$::*HMG1*, *Δhmg2*, *Δjen1*; SAWy644) without mevalonate supplementation in blue light (C) or darkness (D) or with 10 mM mevalonate (MVA) in the media in blue light (E) or darkness (F). The view of *Δhmg2* growth curves is obstructed in (C, E). Basal growth of OptoMEV in the dark (D, F), could be due to leaky expression of *HMG1* from $P_{C120}$. Deletion of *HMG2* is necessary to obtain full optogenetic control of HMG-CoA reductase activity and cell growth. Optical density measurements are reported in Tecan absorbance units (see "Methods"). Biological triplicates were grown in media supplemented with or without 10 mM mevalonate, where mean $OD_{600}$ ($n = 3$) is plotted for each time point, and error bars represent standard deviation. Source data are available online for this figure.

mevalonate importer using OrthoRep. This, however, requires the ability to fine-tune the selection pressure caused by mevalonate deficiency due to the inability to complement a mevalonate auxotroph (i.e., a $\Delta hmg1\Delta hmg2$ mutation) by feeding mevalonate. Following the proof-of-concept experiments with mNeonGreen demonstrating the compatibility of the OptoMEV strain with OrthoRep (Appendix Fig. S1 and Appendix Table S2), we prepared to use OptoMEV to evolve a mevalonate importer from *JEN1*. However, because Jen1p is naturally repressed by glucose-mediated endocytosis and degradation, and glucose is such a preferred substrate in many applications, we introduced a glucose-insensitive mutant of *JEN1*, called *JEN1t* (Appendix Sequence S1), into the p1 of the protoplast-fused strain of OptoMEV (Barata-Antunes et al, 2022). This resulted in OptoRep-*JEN1t*, a strain designed to evolve a mevalonate importer. Next, we searched for an adequate selection pressure for mevalonate importation by measuring the growth of OptoRep-*JEN1t* under different light doses. We reasoned that below a critical light dose threshold, intermediate levels of *HMG1* transcription would result in reduced cell growth, allowing for p1 mutagenesis and sufficient selection pressure to evolve *JEN1t* into a mevalonate importer that would improve cell growth. To dose light, we used light pulsing (where light is applied for a specific amount of time within a total time period), which we have previously shown to be a reliable way to tune the expression of essential genes, and thus cell growth (Zhao et al, 2018). This approach, in the context of OptoRep, allows for the tunability of the selection pressure applied to evolve proteins. We found that incubating the cultures in light pulses of 2% in a forcing period of 100 s (2 s on, 98 s off) and a light intensity of 100 µmol m$^{-2}$ s$^{-1}$ (see "Methods", Fig. EV1), OptoRep-*JEN1t* grows at an intermediate level between full light (close to wild-type growth) and darkness (virtually no growth; Fig. 2A; Appendix Table S3). This represents a level of light-induced *HMG1* expression that is low enough to cause a mevalonate deficiency that limits cell growth and thus imposes a selection pressure to evolve mevalonate importation; while still allowing sufficient cell growth to have OrthoRep continuously mutagenize *JEN1t* in search of mutations that would convert it to a mevalonate importer that would relieve this selection pressure and improve cell growth (Fig. 2B).

Applying this optimal light delivery schedule (2% pulse duration, 100 µmol m$^{-2}$ s$^{-1}$, in 100 s forcing periods) as a semi-permissive condition to evolve OptoRep-*JEN1t*, improved mevalonate-dependent growth. We carried out this evolution experiment in eight independent lineages (SAWy700-1-8; Appendix Fig. S2). Remarkably, after only three passages in semi-permissive conditions, six lineages (Fig. 2C; SAWy700-2,3,5,6,7,8) show an increase in growth, with a significant decrease in lag phase relative to the previous passage (Fig. 2D) and a small but significant increase in growth rate relative to the first mevalonate-supplemented passage (Fig. 2E). Following passaging under semi-permissive light conditions, the different lineages were challenged under full selection pressure by incubating them in complete darkness (resulting in mevalonate auxotrophy), which allowed us to apply purifying selection pressure. Given the significant improvement in mevalonate-dependent growth after only three passages and this purifying selection step, rather than performing additional passages, we proceeded to sequence the p1 plasmid from single colonies derived from each successful lineage. Lineages SAWy700-1 and SAWy700-4 were excluded because they showed either

mevalonate-independent growth (SAWy700-1) or inconsistent growth between passages (SAWy700-4). We found that the *JEN1t* in the three lineages with the highest decrease in lag phase for mevalonate-dependent growth in the third passage (SAWy700-2, SAWy700-3, and SAWy700-5) had mutated, in at least one but not all of the colonies sequenced per lineage, while all colonies from the remaining lineages (SAWy700-6, SAWy700-7, and SAWy700-8) returned only the wild-type *JEN1t* sequence (Fig. EV2A,B).

The same single-point mutation in *JEN1t* is observed in all three independent lineages with this increased mevalonate transport-dependent growth. These lineages exhibit a conserved *JEN1t*$^{Y180C}$ mutation in their p1 plasmid (Fig. EV2C), consisting of a A539G transition mutation (Y180 corresponds to Y273 in the full-length *JEN1* protein). The reason why mutations in *JEN1t* were not observed in the remaining lineages could be partly due to the polyclonal nature of the p1 plasmid, and the low penetrance of the mutant *JEN1t* allele achieved after only three passages (Ravikumar et al, 2018). To test whether the *JEN1t*$^{Y180C}$ mutation is responsible for the increased growth under nonpermissive light conditions with mevalonate supplementation, we expressed it in the OptoMEV strain using a CEN/ARS plasmid. The resulting strain *JEN1t*$^{Y180C}$ grows robustly (at a rate of $0.203 \pm 0.004$ h$^{-1}$) in nonpermissive dark conditions in mevalonate-supplemented medium, but not in media without mevalonate (Appendix Table S4; Fig. EV2D). In contrast, expressing wild-type *JEN1t* does not permit growth in darkness, even with mevalonate supplementation, with similar growth to the strain with an empty vector (Fig. EV2D). To further validate that the *JEN1t*$^{Y180C}$ mutation is responsible for importing mevalonate independently of the OptoMEV background, both isoforms of HMG-CoA reductase ($\Delta hmg1/\Delta hmg2$) were deleted in the strain expressing the *JEN1t*$^{Y180C}$ transporter, which we found to be viable only when supplementing the media with mevalonate (5 mM) and not in media lacking mevalonate (Fig. EV2E). This is consistent with previous studies showing that this double deletion strain is inviable unless rescued by expressing a heterologous HMG-CoA reductase (Basson et al, 1986; Leszczynska et al, 2009), and provides evidence that *JEN1t*$^{Y180C}$ imports mevalonate to the cell. Thus, after only three semi-permissive passages of OptoRep, *JEN1t* evolved into an effective mevalonate importer by a single missense mutation that changed the protein sequence from a tyrosine to a cysteine (Y180C). The fact that the same mutation evolved in three independent OptoRep lineages, and its ability to confer mevalonate-dependent growth in an HMG-CoA reductase-null genetic background, is strong evidence that Y180 is involved in *JEN1* substrate recognition.

## *JEN1t*$^{Y180}$ optimization and mechanism of mevalonate recognition

We hypothesized that *JEN1t*$^{Y180C}$, resulting from a single A539G transition introduced by TPDNAP1, is not necessarily the only or most efficient *JEN1t* mutation at Y180 that can confer a gain of mevalonate importation activity to this transporter. To explore other mutations not as readily accessible to OrthoRep (due to TPDNAP1 biases and the need for multiple mutations), we generated a CEN/ARS *JEN1t* saturation mutagenesis library for Y180 (*JEN1t*$^{Y180NNK}$) that could be used to select for better mevalonate importers. This approach was chosen over constructing individual strains with each substitution not only because it was

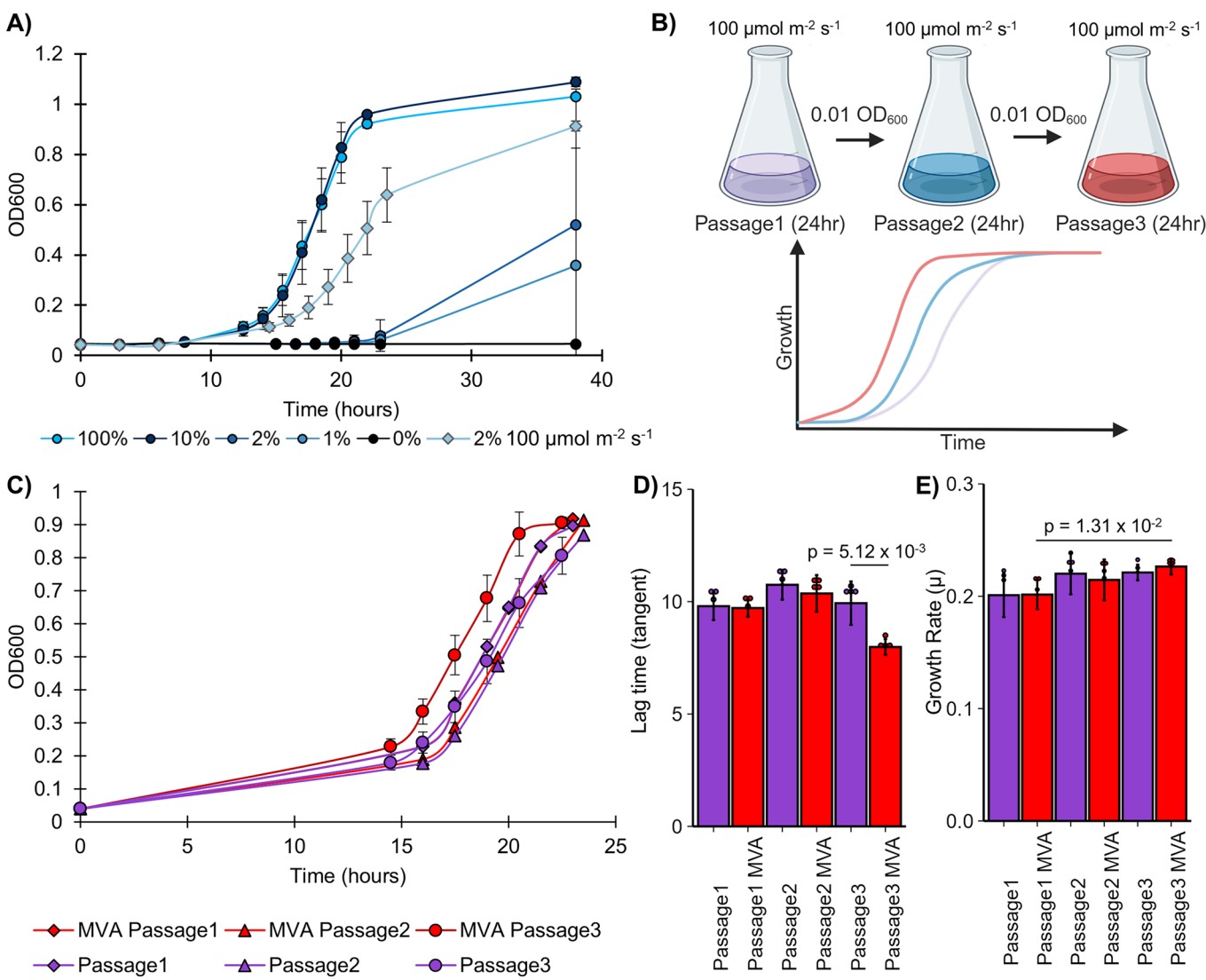

**Figure 2. OptoRep selection yields a gain-of-function *JEN1t* mutant.**

(A) Growth response of OptoRep-*JEN1t* (SAWy700) to varying light pulses of 1% (2 s on/198 s off), 2% (2 s on/98 s off), 10% (10 s on/90 s off), or 100% full light. Starting $OD_{600}$ measurements were 0.01 inoculated from overnight cultures grown under light. Most experiments used a light intensity between 40–60 μmol m$^{-2}$ s$^{-1}$, except one experiment at 2% light pulses that used 100 μmol m$^{-2}$ s$^{-1}$ (labeled in the figure caption), which helped achieve an intermediate growth curve. Data points represent the average of 4 biological replicates, where error bars are the standard deviation. (B) Schematic for OrthoRep passaging, where semi-permissive light conditions of 2% pulses of 100 μmol m$^{-2}$ s$^{-1}$ light intensity result in intermediate levels of growth. After a few passages, OptoRep-*JEN1t* (SAWy700) growth improves, as *JEN1t* acquires the ability to import mevalonate from the medium. (C) Semi-permissive growth curves with or without mevalonate (MVA) of all successful OptoRep-JEN1t lineages (6 of 8 total lineages, two were excluded for reasons described in the text). Data points represent the average of all six successful lineages (SAWy700-2,3,5,6,7,8) with error bars showing the standard deviation for the third passage. Optical density measurements for (A, C) are reported in Tecan absorbance units (see "Methods"). (D) The lag time (calculated through the tangent method) and (E) specific growth rate (μ) obtained from averaged growth curves in (C) are shown for each passage with or without mevalonate (MVA; ($n = 6$ for both (D, E))). Bars represent the average of the six lineages shown in (C–E) with standard error, while points reflect each independent lineage. Statistical comparisons of interest were performed using a two-sided Mann–Whitney U test. Source data are available online for this figure.

more economical and less labor-intensive but also to further demonstrate that our OptoMEV strain can serve as a platform to screen mutant libraries for enhanced growth by exploiting its optogenetic control. We screened 267 library transformants, using 96-well plates, achieving an ~99% probability of sampling the single-point mutation NNK library at full coverage (Nov, 2012). To control for variable inoculation between wells, we measured the ratios of growth obtained in the dark with mevalonate supplementation to the growth obtained without supplementation ($OD_{600}$

in media with 10 mM mevalonate divided by $OD_{600}$ in standard media) in each well after 24 h and then normalized it by the same ratio obtained with a control strain expressing wild-type *JEN1t* (see methods for calculation). Although most colonies showed no growth improvement, several exhibited growth comparable or superior to that of the strain expressing *JEN1t*$^{Y180C}$ (Fig. 3A). The top six colonies showed an average normalized growth ratio of $2.77 \pm 0.09$ compared to $1.82 \pm 0.24$ of the *JEN1t*$^{Y180C}$ control ($P$ value $= 2.79 \times 10^{-3}$). When we sequenced the CEN plasmid in

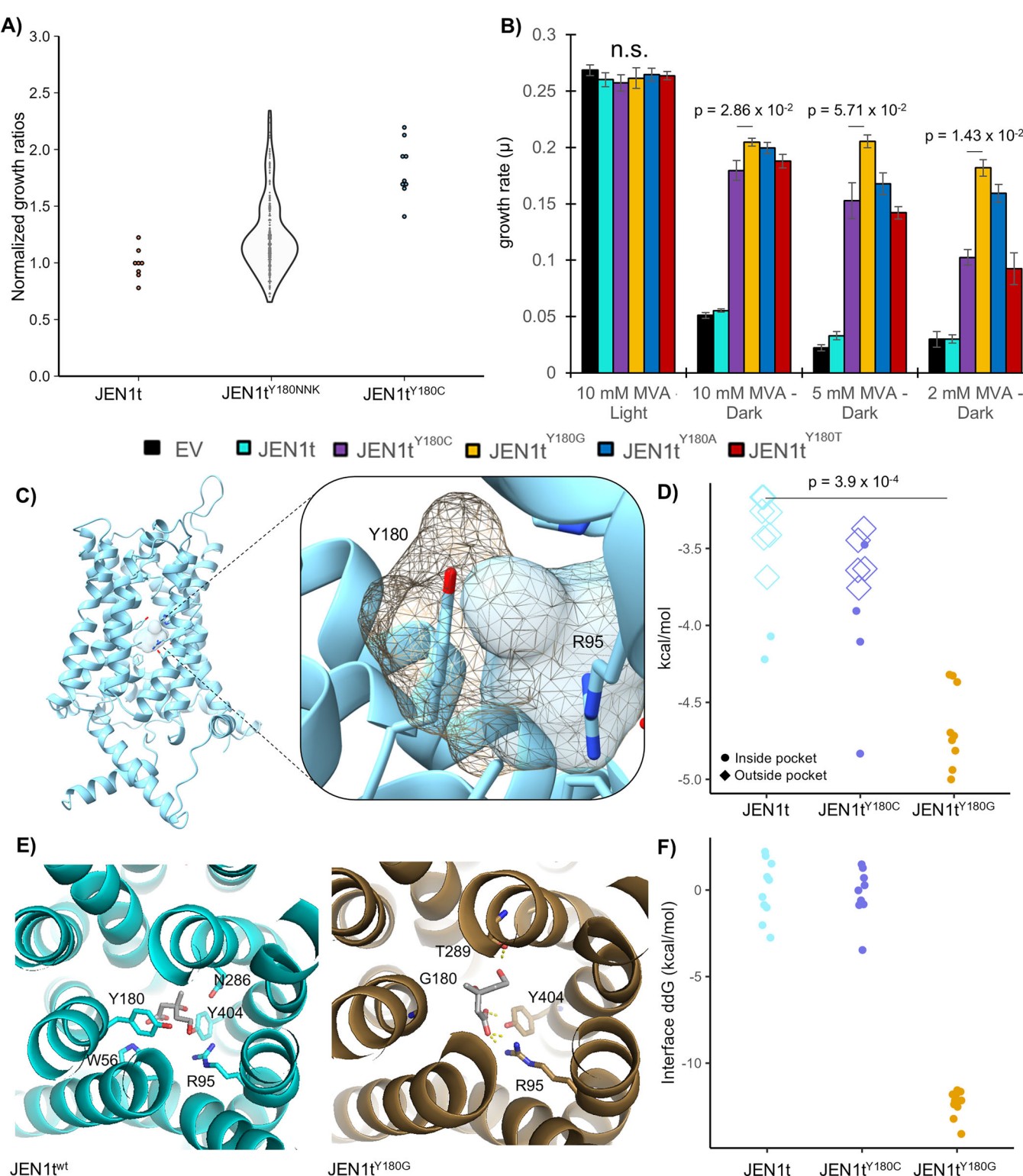

the six top colonies, we found that five of them had Y180 substituted with either glycine or alanine, while one bore a threonine mutation. Interestingly, the Y180C mutation was not isolated from the top six colonies of this less biased screen. Thus, to convert *JEN1t* into a mevalonate importer, there seems to be a preference to substitute the bulky aromatic tyrosine residue at Y180 with a smaller polar (C/T) or nonpolar (G/A) one.

To compare the different *JEN1t* Y180 substitutions isolated from the previous experiments, we assayed their ability to support mevalonate transport-dependent growth. We measured the growth

**Figure 3. JEN1t Y180 mutagenesis and possible mechanism of mevalonate transport.**

(A) Normalized growth ratio of individual transformants derived from the NNK site-saturation mutagenesis library of *JEN1t* at Y180 ($n = 267$). The *Y* axis shows the ratio of $OD_{600}$ measured for each colony after 24 h of growth in the dark with 10 mM mevalonate and the $OD_{600}$ obtained in media without mevalonate, normalized to the same ratio of strains expressing wild-type *JEN1t* (SAWy711; $n = 8$) or the original *JEN1t*$^{Y180C}$ mutation (SAWy712; $n = 9$). (B) Growth rates of *JEN1t*$^{Y180}$ mutants and controls (*JEN1t*$^{WT}$: SAWy711, *JEN1t*$^{Y180C}$: SAWy712, *JEN1t*$^{Y180G}$: SAWy738; *JEN1t*$^{Y180A}$: SAWy739, *JEN1t*$^{Y180T}$: SAWy740, Empty Vector: SAWy705) in the dark when challenged with decreasing mevalonate (MVA) concentrations compared to the maximal growth obtained in full light. Statistical analyses were performed with a two-sided Mann–Whitney *U* test. Only relevant statistical comparisons are shown, with n.s. representing no statistically significant comparisons within the 10 mM MVA light condition. Data bars represent an average of four biological replicates, where error bars are the standard deviation. (C) Molecular structure of Jen1p predicted with AlphaFold. The putative binding pocket lined by the critical R188 (R95 in Jen1p*t*) and Y180 in the wild-type Jen1p*t* (blue surface) is larger in the Jen1p*t*$^{Y180G}$ mutant (gold mesh). (D) Free energies of mevalonate binding obtained from docking mevalonate in the putative binding pocket of the wild-type or Jen1p*t*$^{Y180C/G}$ mutants. Multiple conformational isoforms of mevalonate were generated for each protein ($n = 9$), with the respective binding affinity plotted for each mutant. Individual points are colored according to the general mevalonate binding site: binding inside the putative Jen1p*t* binding pocket (solid circles) or binding at the outer region of the translocation channel (outside pocket; hollow diamonds, Appendix Fig. S3A). Statistics were performed on the binding affinity using a two-sided Mann–Whitney *U* test. (E) Binding configurations of mevalonate in the wild-type (left) or Jen1p*t*$^{Y180G}$ mutant (right) putative binding site, showing key molecular bonds, predicted by Rosetta, to labeled residues (see methods). Models for wild-type Jen1t, Jen1p*t*$^{Y180C}$, and Jen1p*t*$^{Y180G}$ are deposited on ModelArchive under IDs ma-39yos, ma-hnfca, and ma-dqvhp, respectively. (F) Change in the binding free energy (ddG) of mevalonate in the putative binding site of the wild-type and Y180 mutants, calculated with MD simulations using RosettaMP. The top ten molecular simulations for each mutant relative to wild-type are shown. To calculate the free energies, the binding interface was defined as any amino acid residue with an atom within 4 Å of mevalonate, with all intermolecular interactions from these residues contributing to the change in free energy. Source data are available online for this figure.

of strains carrying *JEN1t* with Y180 substitutions incubated under permissive (light) and nonpermissive (dark) conditions and decreasing concentrations of mevalonate in the media (Figs. 3B and EV3). As expected, under permissive light conditions, there are no significant differences in growth between any construct, including all Y180 substitutions, wild-type *JEN1t*, and empty vector (Fig. 3B). Under nonpermissive light conditions and decreasing concentrations of mevalonate in the media we found that strains carrying the *JEN1t*$^{Y180G}$ or *JEN1t*$^{Y180A}$ mutants grow better than those carrying *JEN1t*$^{Y180C}$ or *JENt*$^{Y180T}$ (Figs. 3B and EV3D). Even at the lowest tested mevalonate concentration, the *JEN1t*$^{Y180G}$ substitution can support a robust growth rate of $0.182 \pm 0.007\,h^{-1}$, which is 69.5% of the maximal growth obtained in permissive conditions. These results show that the smaller the residue substituting Y180 is, the better at mevalonate import *JEN1t* becomes.

To gain insight into the molecular mechanism by which mutations at Y180 confer mevalonate substrate recognition to Jen1p*t*, we first mapped this residue onto a three-dimensional AlphaFold model of the full-length protein (Fig. 3C). *JEN1* is a member of the major facilitator superfamily of transporters, with a similar overall structure to other family members, composed of 12 membrane-spanning alpha-helices bundled around a central substrate translocation channel. A model for the transport mechanism of Jen1p has been previously proposed based on the crystal structure of the related major facilitator transporter, GlpTp permease (Soares-Silva et al, 2007, 2011). These studies found that R95 (R188 in the full-length Jen1p) is critical for Jen1p*t* functionality and acts as a hinge facilitating the substrate translocation process (Soares-Silva et al, 2011). Interestingly, Y180 is situated near R95 (at 5.5 Å) on an opposite wall of the substrate channel, in a binding pocket at the core of the translocation channel of the structure predicted by AlphaFold (Fig. 3C). Mutations at this residue alter the shape of the proposed binding pocket, increasing its volume, and potentially allowing greater access to mevalonate.

Mutations at Y180 also seem to alter the apparent preference of mevalonate binding, as predicted by molecular docking. In the wild-type Jen1p*t*, most predicted conformational isoforms of mevalonate preferentially bind outside the translocation channel, at the extracellular mouth of the transporter, probably in a general mechanism of non-specific, mostly electrostatically-driven

interactions of Jen1p*t* to small acids and hydroxyacids (Appendix Fig. S3A). In contrast, the Jen1p*t*$^{Y180C}$ variant shows moderately improved mevalonate binding preference in the translocation channel, in close proximity to R95, while all predicted mevalonate binding conformations in Jen1p*t*$^{Y180G}$ occur in this same translocation site (Appendix Fig. S3A). Consistent with this observation, Jen1p*t*$^{Y180G}$ exhibits a lower mevalonate binding free energy within the translocation site, relative to the wild-type transporter (Fig. 3D). Although promising, it is currently uncertain the degree to which an AlphaFold simulation can serve as a suitable input for docking studies. Therefore, to further support these findings, configurational ensembles relating to the residue binding preference of mevalonate for the wild-type versus mutant Jen1p*t* transporter were generated using the Rosetta macromolecular modeling platform. Although mevalonate is able to bind in the wild-type transporter, it does not interact with R95 but instead forms a hydrogen bond directly with Y180 (Fig. 3E, left). Conversion of this residue to either cysteine or glycine prevents this interaction and instead favors the interaction of mevalonate with R95 (Fig. 3E, right; Appendix Fig. S3B), as well as lowering the free energy of the binding interface (Fig. 3F). Taken together, these findings suggest that Y180 mutations can alter the putative binding pocket of Jen1p; but perhaps more importantly, they may facilitate substrate interactions with R95, a key residue in the translocation process. Given the ability of the *JEN1t*$^{Y180G}$ mutation to recover nearly wild-type levels of growth in nonpermissive conditions with mevalonate supplementation, we reasoned that this evolved transporter was sufficient to explore more complex mevalonate import applications.

## Isoprenoid production enhanced by *JEN1t*$^{Y180G}$ mevalonate import

The availability of a mevalonate importer holds substantial potential for various yeast biotechnological applications, given that mevalonate is a precursor of the vast family of isoprenoid natural products. As proof-of-concept, we expressed *JEN1t*-derived mevalonate importers in a strain engineered to produce farnesene, a sustainable aviation fuel, to test whether it could produce farnesene from mevalonate supplemented in the media. These strains were engineered to overexpress the lower mevalonate pathway (comprising *ERG12*, *ERG8*, and *ERG19*), a stabilized farnesene synthase

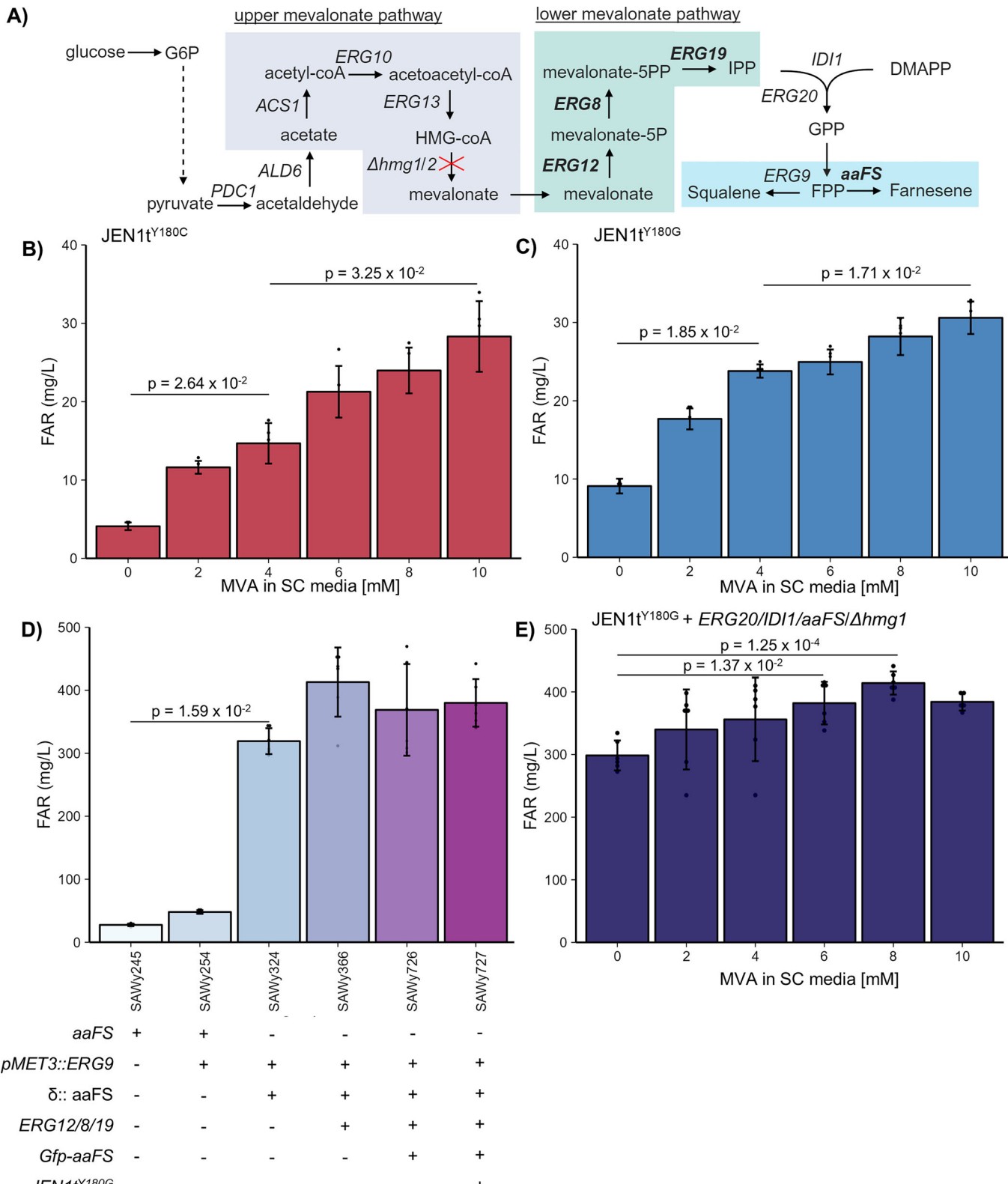

◄ **Figure 4. Effect of expressing *JEN1t* variants on farnesene production.**

(A) Farnesene biosynthesis comprises an upper and lower pathway. The upper mevalonate pathway can be blocked by the double Δhmg1/Δhmg2 deletion, disabling endogenous mevalonate synthesis. Farnesene production can be achieved by overexpressing the lower mevalonate pathway (shown in bold) and importing mevalonate from the medium. (B, C) Farnesene production in Δhmg1/Δhmg2 strains overexpressing the lower mevalonate pathway and *JEN1t*$^{Y180C}$ (SAWy717) (B) or *JEN1t*$^{Y180G}$ (SAWy722) (C), fermented in media containing different concentrations of mevalonate (MVA). Double Δhmg1/Δhmg2 deletion strains were outgrown in medium containing 2 mM mevalonate starting from a 0.1 starting inoculation for 24 h. The following outgrowth strains were washed and resuspended in fresh medium containing the specified concentration of mevalonate and fermentation for 48 h. (D) Farnesene production in strains containing different genetic elements without adding mevalonate to the media: SAWy245 (aaFS), SAWy254 (aaFS, P$_{MET3}$::ERG9), SAWy324 (δ::aaFS, P$_{MET3}$::ERG9), SAWy366 (SAWy324 + lower MVA pathway (ERG12, ERG8, ERG19)), SAWy726 (SAWy366, GFP-aaFS), SAWy727 (SAWy366, GFP-aaFS, *JEN1t*$^{Y180G}$). Multi-copy integration of the farnesene synthase into yeast delta (δ:: aaFS) sites was performed in SAWy324 and carried into subsequent daughter strains where specified. (E) Farnesene production by feeding different concentrations of mevalonate (MVA) to a strain containing an additional copy of *ERG20*, *IDI1*, aaFS, and a Δhmg1 deletion background (SAWy749). Fermentations were performed with mevalonate feeding and *ERG9* inhibition. Average and individual data points for 5–6 biologically independent replicates are shown with error bars showing the standard deviation. Data analysis was performed by Kruskal–Wallis with Dunn's post hoc comparisons for all panels. Only relevant statistical comparisons are shown. Metabolite abbreviations: glucose-6-phosphate (G6P), mevalonate-5-phosphate (mevalonate-5P), mevalonate-3,5-pyrophosphate (mevalonate-5PP), geranyl pyrophosphate (GPP), farnesyl pyrophosphate (FPP). Source data are available online for this figure.

from *Artemisia annua* (GFP-aaFS; Fig. 4A), and either *JEN1t*$^{Y180C}$ (SAWy717) or *JEN1t*$^{Y180G}$ (SAWy722; Cheah et al, 2023). However, to ensure that farnesene production was dependent on mevalonate importation, we also deleted Δhmg1/Δhmg2, which prevents endogenous mevalonate synthesis.

As expected, expression of *JEN1t*$^{Y180C}$ or *JEN1t*$^{Y180G}$ allows the production of farnesene from exogenous mevalonate. At higher mevalonate concentrations (10 mM), there is no significant difference in farnesene production between strains expressing either mevalonate importer (Fig. 4B,C). However, at lower mevalonate concentrations (2, 4 mM) the farnesene production of the strain containing *JEN1t*$^{Y180G}$ is significantly higher than that of the strain containing *JEN1t*$^{Y180C}$ (Fig. 4B,C; P values = $8.04 \times 10^{-3}$). This suggests that both transporters have similar importation rates, but that *JEN1t*$^{Y180G}$ has a higher affinity for mevalonate. Farnesene production without mevalonate supplementation is likely due to the carryover of mevalonate from the growth phase (where mevalonate was present at 2 mM) into fermentation. Interestingly, expression of either transporter in a strain capable of synthesizing mevalonate endogenously (*HMG1*, *HMG2*) also boosts farnesene production when mevalonate is supplemented in the media (Appendix Fig. S4). These results demonstrate that these importers can supplement mevalonate to strains engineered to produce isoprenoids to enhance their production.

These effects, however, were observed in strains with relatively low farnesene production, which made us wonder whether a similar effect would be apparent in strains engineered with a higher metabolic flux for farnesene production. To explore this possibility, we expressed a mevalonate importer in a strain in which *ERG9* is transcriptionally repressed by methionine, using the P$_{MET3}$ promoter (P$_{MET3}$-*ERG9*), which has been previously shown to significantly enhance isoprenoid production in engineered strains (Ro et al, 2006). Having additional copies of farnesene synthase along with overexpressing the lower mevalonate pathway results in a strain (SAWy366) that produces 413 ± 50.1 mg/L of farnesene, a ~15-fold increase over the starting strain (SAWy245), and the production seen in the initial feeding experiments (Fig. 4D). Expressing *JEN1t*$^{Y180G}$ (SAWy727) does not result in a significant effect from mevalonate supplementation (Appendix Fig. S5A). We first ruled out the possibility of having a problem of *JEN1t*$^{Y180G}$ expression or membrane localization in this strain, using a GFP-tagged version of the protein (Appendix Fig. S6). When we reduced the metabolic flux by de-repressing *ERG9* (not adding methionine

to the media) we found that mevalonate supplementation resulted in a small but significant increase in farnesene production (Appendix Fig. S5B), showing that *JEN1t*$^{Y180G}$ is active in this strain. This led us to consider the possibility that *JEN1t*$^{Y180G}$ importation was limited by an uphill mevalonate gradient due to pathway upregulation caused by *ERG9* repression (Özaydin et al, 2013; Wegner et al, 2021). This has been observed with the functionally related mammalian monocarboxylate transporter *MCT1*, whose transport direction is influenced to a degree by the substrate gradient (Wang et al, 2024).

To test this possibility, we aimed to reduce the endogenous mevalonate synthesis by deleting only the major HMG-coA reductase isoform *HMG1*. The remaining *HMG2* in this strain (SAWy749) was able to carry most but not all of the original flux to produce 298.497 ± 23.841 mg/L farnesene (Fig. 4E), still approximately tenfold higher than SAWy245. We found that expressing *JEN1t*$^{Y180G}$ in this strain improves farnesene production to 414.10 ± 18.49 mg/L when supplementing the methionine-containing medium with 8 mM mevalonate. This 39% increase in production shows that mevalonate importation can increase the production of an isoprenoid, even at higher metabolic fluxes.

## Farnesene production by a *JEN1t*$^{Y180G}$-mediated *S. cerevisiae* consortium

Mevalonate is an expensive substrate, so a more realistic scenario where mevalonate importation is used to boost isoprenoid production is in the context of microbial consortia, in which one member uptakes the mevalonate secreted by another. Previous studies have shown that yeast has the natural ability to secrete mevalonate when produced in excess, which is irreversible and mediated by multiple redundant transporters (Rodriguez et al, 2016; Wegner et al, 2021; Wegner and Avalos, 2024). This phenomenon should allow consortia in which mevalonate secreted by one strain is converted to a product of interest by another strain that uptakes the secreted mevalonate through a neofunctionalized Jen1t mevalonate importer. As a proof-of-concept for such application, we established a yeast–yeast consortium in which an upstream member secretes excess mevalonate, a second downstream member imports it through *JEN1t*$^{Y180G}$, and both are engineered to convert mevalonate to farnesene (Fig. 5A,B). Such consortia would enable the recovery of some of the mevalonate lost as byproduct by the upstream strain—which cannot reimport it due

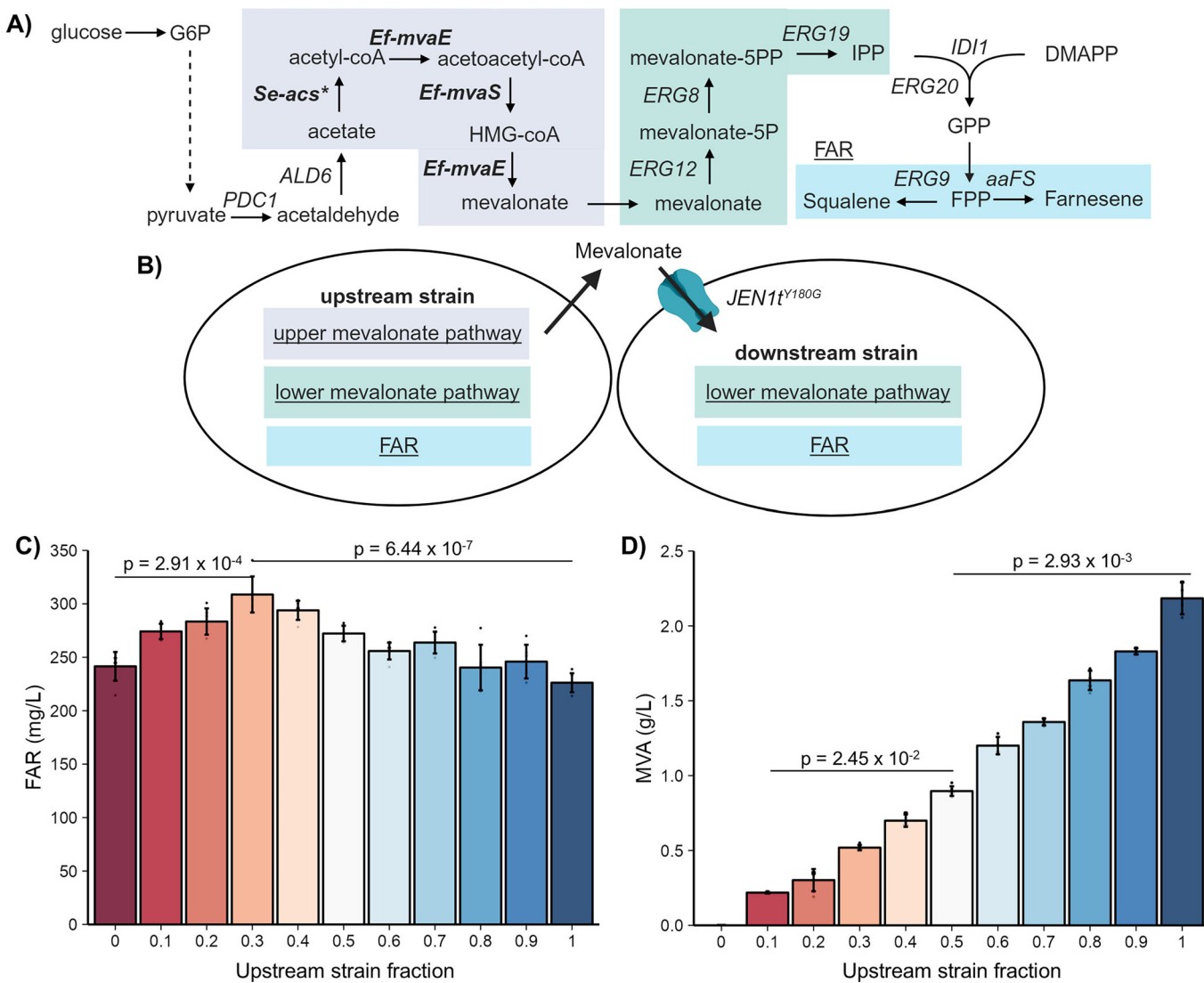

**Figure 5. Farnesene production in consortia of yeast strains exchanging mevalonate through *JEN1t*$^{Y180G}$-enabled mevalonate importation.**

(A) The upstream strain (SAWy733) was engineered to overproduce mevalonate by overexpressing three heterologous enzymes (bold). *Se-acs** denotes the L641P feedback inhibition-resistant acetyl-CoA synthetase from *Salmonella enterica*. Additional mevalonate pathway enzymes (comprised by *ERG12*, *ERG8*, and *ERG19*) and the farnesene (FAR) production module (comprised by aaFS and methionine-induced repression of *ERG9*), were overexpressed in both the upstream and downstream strains (SAWy749) of the consortium. In addition, *IDI1* and *ERG20* were overexpressed in the downstream strain. Solid arrows represent individual enzymatic steps, while dashed arrows represent multiple enzymatic reactions. (B) Schematic of the mevalonate-exchanging consortium for farnesene production, comprised of an upstream strain (left), expressing all three modules for farnesene production; and a downstream strain (right), expressing *JEN1t*$^{Y180G}$ instead of the upstream mevalonate pathway. (C, D) High cell-density fermentations of mevalonate-exchanging consortia for farnesene production. Charts show the production (after 48 h) of (C) farnesene (FAR), and (D) mevalonate (MVA), depending on the starting consortia composition expressed, as the fraction of the upstream strain (SAWy733). The upstream and downstream strains were cultivated separately for 24 h prior to mixing and fermentation. Average and individual data points for 6 biologically independent replicates are shown with error bars showing the standard deviation. Data analysis was performed by Kruskal–Wallis with Dunn's post hoc comparisons for all panels. Source data are available online for this figure.

to the uphill gradient—by using a downstream strain that can import it, thanks to not having such a gradient, and convert it to a final product.

The upstream strain (SAWy733) is engineered with the upper- and lower mevalonate pathways, as well as a farnesene (FAR) module comprised of aaFS and P$_{MET3}$-*ERG9*, while the downstream strain (SAWy749) only has the lower mevalonate pathway (as well as Δ*hmg1* to prevent a mevalonate uphill gradient) and FAR but also expresses *JEN1t*$^{Y180G}$ (Fig. 5B). Cultivated separately, both

strains have similar farnesene production, with the upstream strain producing $226.20 \pm 18.86$ mg/L and the downstream strain $241.51 \pm 13.45$ mg/L. However, when mixing the strains in high cell-density fermentations at different ratios, in which the fraction of upstream strain varied from 0.1 to 0.9 in 0.1 increments, we found that the amount of farnesene produced by the consortia increased to a maximum of $308.72 \pm 16.87$ mg/L at a 0.3 fraction of the upstream strain, which is significantly higher than the individual production of either strain by ~32% (Fig. 5C). As

expected, the final concentrations of mevalonate byproduct also varied depending on the fraction of upstream strain, ranging from $0.22 \pm 0.01$ g/L (at 0.1) to $2.18 \pm 0.11$ g/L (Fig. 5D). By pre-growing each strain separately and then mixing them in precisely measured proportions for high cell-density co-culture fermentations, we eliminate any effects caused by growth differences or growth competition between the strains. This is further validated by the minimal drift in consortia composition measured at the end of the fermentation (Fig. EV4). This proof-of-concept demonstrates how mevalonate importation, enabled by $JEN1t^{Y180G}$ can be useful to improve the production of an isoprenoid in yeast consortia.

## Discussion

Here, we establish OptoRep, an optogenetic strategy to fine-tune the selection pressure applied in an OrthoRep evolutionary campaign, which enabled the evolution of a gain-of-function mutation in $JEN1$ that neofunctionalizes it for mevalonate transport. With OptoRep, microbial growth can be constrained to an essential cell function, similar to standard OrthoRep, but without needing to supplement the medium with nutrients or metabolites to compensate for deficits in that essential function. Instead, in OptoRep, light is used to fine-tune the expression level of a gene that performs that essential function—in this demonstration, $HMG1$ to synthesize mevalonate. This allows cell growth under semi-permissive light conditions in which the essential function is limiting and thus imposing a selection pressure, but still enabling sufficient growth for OrthoRep mutagenesis to drive the evolution of a complementary function in the gene contained in the p1 plasmid—in this demonstration, mevalonate import in $JEN1t$. In conventional OrthoRep, the selection pressure can be increased after a few passages by changing the medium composition (e.g., reducing the concentration of an essential metabolite or increasing that of a growth inhibitor). In OptoRep, the selection pressure may be increased by reducing the light dose during cell growth, which in turn reduces the expression of the essential gene, and increases the pressure to grow by the evolved complementary activity on the p1-contained gene. Semi-permissive passaging also preserves the rapid rate of mutagenesis previously reported for OrthoRep, as we found that genetic and phenotypic changes appeared after only three passages (Ravikumar et al, 2018; Rix et al, 2020). Moreover, optogenetics provides a high level of flexibility in applying user-defined selection setpoints to enable genetic drift (semi-permissive or permissive light conditions) or to apply purifying selection (dark conditions) through simple adjustments in light exposure. Finally, OptoRep is compatible with existing automated in vivo evolutionary platforms (such as eVOLVER), with many examples of autonomous optogenetic control of microbial growth (Carrasco-López et al, 2020; Zhong et al, 2020). One potential limitation of this study is that, because the Y180C mutation consistently emerged relatively quickly compared to other OrthoRep studies, we were unable to measure the long-term stability of the $P_{C120}$-$HMG1$ construct in evolutionary campaigns longer than the three passages (or ~72 h) that took for this mutation to emerge. Therefore, the stability of $P_{C120}$-controlled constructs in projects that need additional passages (e.g., that require the accumulation of multiple mutations) remains to be determined. While in previous studies we have observed a relatively long stability of essential genes

(such as $PDC1$ in glucose) controlled by $P_{C120}$ or derived circuits, this stability is liable to vary depending on specific features of the system and the light conditions used to tune the selective pressure in the evolutionary campaign (Lalwani et al, 2021; Zhao et al, 2018, 2020, 2021). Nevertheless, as shown here, OptoRep can serve as a suitable starting point for the mutagenesis of a target. Thus, OptoRep maintains the advantages of traditional OrthoRep while allowing new targets to be accessed with highly flexible optogenetic selection.

OptoRep can facilitate the emergence of neofunctionalizing gain-of-function mutations, which may be difficult to obtain with conventional OrthoRep. As shown here, natural mevalonate uptake in $S.\ cerevisiae$ is negligible and independent of the wild-type $JEN1$ or $JEN1t$. This made what would be the standard OrthoRep approach to improve mevalonate importation by gradual reduction of mevalonate media supplementation impractical. However, OptoRep circumvents this standard requirement by controlling the expression of a gene or pathway whose function is intended to be replaced by an evolved function in the p1-encoded gene of interest. This approach allowed the recovery of a primary gain-of-function mutation that neofunctionalizes $JEN1t$. We were surprised to find that only a single-point mutation was enough to confer a new function to $JEN1t$, which, in retrospect, might have been possible to recover using standard mutagenesis methods, such as an error-prone PCR library of $JEN1t$. However, this does not detract from the advantage of OptoRep to tune the selection pressure in evolutionary campaigns, which might prove essential for proteins that require more complex multi-mutational pathways for neo-functionalization. Two previous studies have used OrthoRep for gene neofunctionalization. In one of them, a secondary promiscuous activity was evolved in TrpB, allowing it to utilize new indole derivatives as substrates (Karapanagioti et al, 2024; Rix et al, 2020). In a more recent study, the substrate profile of amino acid-polyamine-organocation transporters was expanded to allow recognition of new substrates (Karapanagioti et al, 2024; Rix et al, 2020). Similarly, we used OrthoRep to find a gain-of-function mutation that neofunctionalizes $JEN1t$ into a mevalonate importer, but only made possible by the light-tunability of $HMG1$ expression afforded by OptoRep.

Our results revealed that Y180 is a critical residue for Jen1p substrate selectivity, and specifically for gain-of-function mutations that neofunctionalize $JEN1t$ into a mevalonate transporter. It is remarkable that in three independent OptoRep evolutionary lineages, the one and only mutation that emerged was Y180C. This is a strong indication that Y180 is a key residue in the substrate selectivity of Jen1p. All mutations were constrained to a single nucleotide transition (TAC to TGC), resulting in the key cysteine substitution, which may be partly due to the mutation spectra bias of OrthoRep. To evolve the better mevalonate transporter $JEN1t^{Y180G}$, an additional transversion mutation would be required (GGC), which TPDNAP1-4-2 introduces at a much lower frequency than transitions (this substitution bias has been recently reduced with the development of new TPDNAP1 variants, which could have helped find the $JEN1t^{Y180G}$ mutation) (Ravikumar et al, 2018; Rix et al, 2024). However, because the $JEN1t^{Y180C}$ mutation is clearly sufficient to confer the mevalonate importation activity to $JEN1t$ needed for improved cell growth in semi-permissive light conditions, and this mutation is more readily accessible to OrthoRep than $JEN1t^{Y180G}$ this later mutation did not

emerge in our OptoRep experiments. While increased selection stringency (achieved by lowering mevalonate concentrations) and additional passaging might have eventually recovered this mutation, our OptoRep strategy is ideally suited for scenarios in which the primary challenge is undetectable activity in the targeted protein. This approach can effectively identify promising neofunctionalized variants that can be further optimized by conventional OrthoRep or other methods.

The key role of Y180 in substrate recognition is supported by the structural comparison of Jen1p to other functionally related monocarboxylate transporters. The cryo-EM structure of the MCT1 (a human monocarboxylate transporter) bound to lactate revealed the importance of R313 in lactate recognition as a substrate (Wang et al, 2021). Similarly, an arginine residue, R95 (R188 in the full-length transporter), in Jen1pt has been shown to be essential for function (Soares-Silva et al, 2011). In the structure of Jen1p predicted by AlphaFold, the Y180 residue lies opposite R95 in the proposed lactate-binding pocket (Fig. 3E). According to our MD simulations, mutations of Y180 to cysteine or glycine enlarge the substrate-binding pocket and favor the interaction of R95 with the carboxylic acid functional group of mevalonate, closely resembling the interaction of R313 with the carboxylic acid of lactate observed in the structure of MCT1 bound to this substrate, which was proposed to be essential for lactate transport (N. Wang et al, 2021). This mechanism is further supported by the ability of MCT1 to recognize mevalonate as a substrate when bearing the F360C mutation near this critical arginine residue, resembling the Y180C mutation in Jen1pt. Finally, the fact that the $JEN1t^{Y180C}$ mutation is not predicted to dramatically improve the mevalonate binding affinity (Fig. 3D,F) suggests that the alternative mevalonate binding conformation and new interaction with R35, enabled by the mutation, are mostly responsible for the gain of mevalonate transport in $JEN1t$.

OptoRep may facilitate the in vivo continuous evolution of other transmembrane metabolite transporters, which play a key role in microbial metabolic engineering. For example, monoterpenes (Dunlop et al, 2011), sesquiterpenes (Zhang et al, 2016), and fatty and fusel alcohol production (Ahmed et al, 2021; Hu et al, 2018) have benefited from the overexpression of efflux transporters to boost titers and reduce toxicity. However, engineering the substrate specificity of efflux transporters to export products or intermediate metabolites (the latter for consortia applications), is a largely untapped strategy, with a notable example being the evolution of acrB from *E. coli* to improve n-butanol production (Ahmed et al, 2021; Fisher et al, 2014). In this study, we showed that $JEN1t^{Y180G}$ enabled microbial consortia to improve farnesene production by the exchange of mevalonate between strains, which has not been previously shown. $JEN1t^{Y180G}$ could equally be applied in microbial consortia developed for the production of, in principle, any other isoprenoid, which make up a vast family of valuable natural products (Navale et al, 2021; Vickers et al, 2017). Furthermore, OptoRep could also be applied to develop transporters of other intracellular metabolites without known transporters, simply by controlling the endogenous biosynthesis of the metabolite of interest with light, and evolving gain-of-function mutations that neofunctionalize transporters of molecules with some similarities to the metabolite. Selection could be performed by linking transport to cellular fitness. Developing new transporters for intermediate metabolites would not only allow new designs of microbial

consortia for chemical production, but also potentially enable new biotechnologies, for example, cell biosensors in which engineered cells assimilate metabolites they do not naturally import from the extracellular environment.

OptoRep can also be used to engineer intracellular pathways and enzymes. For example, our existing OptoMEV strain could be used to optimize the bacterial DXP/MEP pathway in yeast, which is a more efficient isoprenoid production pathway and has been functionalized in yeast but under low aerobic conditions (Kirby et al, 2016). The DXP pathway enzymes IspG/H mediate oxygen-sensitive iron-sulfur reduction reactions, which could be evolved to function more effectively under aerobic conditions in yeast using OptoRep to evolve OptoMEV to grow in the dark (Kirby et al, 2016; Partow et al, 2012). In addition, OptoMEV could be used to evolve extremophilic enzymes from the archaeal mevalonate pathway, which are immune to yeast regulatory inhibition (Kazieva et al, 2017; Primak et al, 2011), and adapt them to mesophilic conditions compatible with yeast. Other light-conditional auxotrophic strains could be developed, similar to OptoMEV, but for other key metabolites, which would allow the in vivo continuous evolution of enzymes or pathways that complement the auxotrophy using OptoRep. For example, to develop alternative pathways for cytosolic acetyl-CoA production (van Rossum et al, 2016), or optimize synthetic pathways such as non-oxidative glycolysis (Bogorad et al, 2013). Thus, OptoRep is a versatile strategy to evolve new or improved protein functions that require fine-tuning of the selection pressure, offering great potential to optimize metabolic pathways, enzymes, or transporters.

# Methods

**Reagents and tools table**

| Reagent/resource | Reference or source | Identifier or catalog number |
|---|---|---|
| **Experimental models** | | |
| *E. coli (DH5a)* | NEB | C2987I |
| *S. cerevisiae* | This study | Appendix Table S6 |
| **Recombinant DNA** | | |
| Plasmids | This study | Appendix Table S5 |
| **Oligonucleotides and other sequence-based reagents** | | |
| Primers | This study | Appendix Table S7 |
| **Chemicals, enzymes, and other reagents** | | |
| CloneAmp HiFi polymerase | Takara | 639298 |
| Phire Plant PCR mix | Thermo | F160S |
| T4 Ligase | NEB | M0202S |
| XmaI | NEB | R0180S |
| AscI | NEB | R0558S |
| MreI | Thermo | ER2021 |
| NotI | NEB | R0189S |
| XhoI | NEB | R0146S |

| Reagent/resource | Reference or source | Identifier or catalog number |
|---|---|---|
| DpnI | NEB | R0176S |
| Miniprep kit | Epoch Life Science | 2160-050 |
| E.N.Z.A. gel extraction kit | Omega Bio-tek | D2500-00 |
| LB broth | Sigma | L3522 |
| BD Difco yeast nitrogen base without amino acids | Fisher | BD 291920 |
| D-(+)-Glucose | Sigma | G7021 |
| Myo-inositol | Sigma | I5125 |
| Ammonium sulfate | Sigma | A4418 |
| L-glutamic acid monosodium salt hydrate | Sigma | G1626 |
| Alanine, arginine, asparagine, aspartic acid, cysteine, glutamic acid, glutamine, glycine, histidine, isoleucine, leucine, lysine, methionine, phenylalanine, proline, serine, threonine, tryptophan, tyrosine, and valine | Sigma-Aldrich | N/A |
| BD Difco Agar | Fisher | DF0812 |
| Potassium Chloride | Sigma | P3911 |
| Potassium Hydroxide | Sigma | 221473 |
| Hydrochloric Acid | Sigma | 320331 |
| Ampicillin | Sigma | A9393 |
| Zeocin | Thermo | R25005 |
| Geneticin (G418) | Thermo | 10131035 |
| CloneNat (Nourseothricin) | RPI | N51200 |
| 24-well plate | Thermo | 142475 |
| 96-well glass-bottom plate | Greiner | 655892 |
| Concavalin A | Sigma | C5275 |
| MemBrite™ Fix 640/660 | Biotium | 30097 |
| 5-FOA | Sigma | F5013 |
| Mevalonolactone | Fisher | AC428300010 |
| Farnesene | Amyris | N/A |
| Dodecane | Sigma | 297879 |
| **Software** | | |
| R | https://www.r-project.org/ | |
| ImageJ | https://imagej.net/ij/ | |
| PyRx | https://pyrx.sourceforge.io/ | |
| Rosetta | https://rosettacommons.org/ | |
| **Other** | | |
| Infinite F Plex reader | Tecan | |
| W1 SoRa | Nikon | |
| Eclipse TI | Nikon | |
| Quantum meter | Apogee | MQ-510 |

| Reagent/resource | Reference or source | Identifier or catalog number |
|---|---|---|
| Light panel | HQRP | |
| Timer Switch | NearPow | |
| 1260 Infinity HPLC | Agilent | |
| Aminex HPX-87H | Bio-Rad | 1250140 |
| 7890B GC | Agilent | |
| DB-Wax UI (30 m length, 0.25 mm diameter, 0.5 μm film) | Agilent | 122-7033UI |

## Plasmid construction

All plasmids not provided by other labs were derived from the pJLA series of vectors, including the CEN/ARS vector with replacement of the 2 μ plasmid origin and all vectors used for integration (targeting the YARCdelta5, *LEU2*, *URA3*, *TRP1* and *HIS3* locus (Zhao et al, 2018, 2020). Insertion of relevant genes and promoter/terminator pairs was performed using previously established restriction enzyme pairs and T4 ligation using enzymes obtained by NEB (Avalos et al, 2013). Alternatively, regions of interest were PCR amplified using Takara CloneAmp HiFi polymerase and constructed using Gibson assembly (Gibson et al, 2009). Relevant enzymes were obtained from existing plasmids (mvaE, mvaS, se-ACS (L461P)), the yeast genome (*JEN1*, *ERG12/8/19/20*, *IDI1*), or were synthesized separately (Wegner et al, 2021). The *Artimisia annua* farnesene synthase variant Aa_FS_C was synthesized by BioBasic and codon optimized for yeast expression (Meadows et al, 2016). The previously reported *JEN1ΔNT94ΔCT33* was amplified directly from the yeast genome as a truncated ORF (supplementary sequence 1; Barata-Antunes et al, 2022). OrthoRep plasmids were obtained from Dr. Chang Liu: the p1-integration cassette (GA-Ec78) and the wild-type (AR-Ec318) or TPDNAP1-4-2 error-prone variant (AR-Ec633; Ravikumar et al, 2018). All plasmids used in the current study are listed in Appendix Table S5. Epoch Life Science Miniprep, Omega Gel extraction, and PCR purification kits were used to purify plasmids and DNA. All vectors were sequenced with Sanger sequencing (GENEWIZ), or nanopore sequencing (Plasmidsaurus) prior to introduction into yeast. Plasmids were cloned and propagated in chemically competent DH5α strains using LB media at 37 °C supplemented with 100 μg/ml ampicillin (Inoue et al, 1990).

## Yeast medium and transformation

Synthetic complete (SC) media was used for all yeast growth, transformation, and fermentation unless otherwise stated. Synthetic complete media is composed of 1.5 g Difco yeast nitrogen base without amino acids and ammonium sulfate, 5 g ammonium sulfate, 20 g glucose, 200 μM inositol, and 2 g amino acid powder mix per liter. Individual amino acids were dropped out of the amino acid powder mix where relevant to maintain CEN/ARS plasmid selection or select for auxotroph-based homologous recombination based on the relevant locus of integration. Yeast transformation was performed using the Lithium acetate procedure following established methods, where integration plasmids were

linearized with restriction enzymes prior to transformation (Gietz and Woods, 2002; Inoue et al, 1990). The constructed yeast strains are listed in Appendix Table S6. Integration into the p1 plasmid was performed similarly to integration into standard loci; however, following transformation colonies were restreaked twice onto selective plates (SC-URA) to ensure enrichment of integrated p1 plasmid, as has been previously described (Ravikumar et al, 2018). For delta-integration SC medium containing L-glutamic acid monosodium as a substitute nitrogen source was used for Zeocin selection (2000 µg/ml). Colony screening of delta-integration transformants for farnesene production was performed as previously described using high cell-density fermentation, with addition of a 10% v/v dodecane overlay (Wegner et al, 2021). Other integrations, including integration into the p1 plasmid, were verified by PCR genotyping with colony screening by fermentation, where appropriate. Gene deletions were performed using homology-based methods with either G418 (Geneticin) selection for *HMG2* deletion or nourseothricin (ClonNat) for *HMG1* or *JEN1* deletion. The primers used for gene deletions are listed in Appendix Table S7. For deletion of the *HMG1* in strains already lacking the *HMG2* paralog, 10 mM mevalonate was supplemented into the medium. Protoplast fusion was performed as described previously with one modification: strains were plated on selective agar containing 0.6 M KCl and covered with a 4% agar overlay rather than direct agar embedding (Javanpour and Liu, 2019).

The $P_{MET3}$::*ERG9* strains were constructed using our previously established plasmids and strains using marker-less CRISPR integration (Wegner et al, 2021). The repair construct was amplified from JCY48-A6 and inserted into a pUC57 vector, following which the majority of the HIS marker used for selection was removed using backbone PCR and isothermal assembly (the HIS-terminator was spared to prevent upstream transcription bleed-through). The same Cas9 and sgRNA sequence was used (pV1382_ERG9_1) to introduce double-stranded DNA breakage. The Cas9 plasmid was co-transformed with the repair DNA onto media lacking methionine; following transformation strains were verified using genotyping and Sanger sequencing and the Cas9 plasmid was cured using 5-FOA selection (Boeke et al, 1987). The OptoMEV strain was similarly constructed. The $P_{C120}$ promoter (Zhao et al, 2018) was used to replace the $P_{MET3}$ promoter in the $P_{MET3}$::*ERG9* repair plasmid. Homology to the *HMG1* locus was introduced using primers targeting the putative *HMG1* promoter. The pV1382 plasmid (pV1382_HMG1) was again used; however, the sgRNA sequence was modified to target the native promoter of *HGM1* (AGCATGCCGCTGCTATTCAA). The following transformation strains were first phenotypically validated for light-dependent growth, then subjected to genotyping and plasmid removal.

## Microscopy

For microscopic analysis, yeast cells were grown overnight from single colonies in synthetic complete media. Cells were then back-diluted by a factor of 1:100–1:2000 and outgrown for 16 h. The dilution which reached mid-exponential phase (between $OD_{600}$ 3-5) was then used for imaging. Cells were diluted to 0.1 $OD_{600}$ and adhered to a 96-well glass-bottom plate using 1 mg/ml concanavalin A. Images were captured using an upright epi-fluorescent Nikon Eclipse TI with a 100X oil objective. Quantification of peripheral versus internal fluorescence was performed as described previously (Barata-Antunes et al, 2022).

## Optogenetic growth and mevalonate feeding

For characterization of growth and growth rate biologically independent transformants were grown overnight (~16 h) in full blue light (40–60 µmol m$^{-2}$ s$^{-1}$ as measured by an Apogee quantum meter (MQ-510)), delivered by a blue light-emitting diode (LED) panel (HQRP 12-inch blue LED 14 W) unless otherwise specified. Where these standard light conditions not used, the light intensity was as specified with the pulsing controlled by a NearPow timer switch outlet using the specified pulse duration; for example, a 2% (2 s/100 s) light condition would reflect a 2 s stimulus delivery followed by 98 s of darkness. Following overnight growth strains were diluted 1:1 and normalized to ~1.0 $OD_{600}$ (spectrophotometer units) prior to back-dilution for outgrowth. Outgrowth assays were performed starting from a 0.01 starting $OD_{600}$ in 1 ml media (as defined below), in a 24-well plate with 30 °C incubation and 200 RPM shaking, and appropriate light conditions or covered with aluminum foil.

To determine the optimal selection conditions, we explored how light preconditioning (overnight light conditions prior to the experiment), starting inoculation, and light pulse duration interact to affect optogenetic growth. Initial tests indicated that low pulse durations, such as 10% light in 100 s forcing periods (10 s stimulus out of 100 s total) and intensities between 40 and 60 µmol m$^{-2}$ s$^{-1}$, lead to full growth under all tested conditions. Despite this observed light saturation, commonly used short pulse durations (2% light) do not sufficiently rescue growth (Fig. EV1A–D). To further optimize growth recovery, the light intensity was increased to 100 µmol m$^{-2}$ s$^{-1}$. This higher intensity results in significantly higher levels of growth relative to 40–60 µmol m$^{-2}$ s$^{-1}$. For the 0.01 overnight pulsed inoculation this increase in light intensity results in an approximately sevenfold increase in growth rate relative to nonpermissive conditions, while still applying selection by only recovering 73.7 ± 10.5% of the full growth rate predicted by permissive conditions (Fig. 2A; Appendix Table S3).

Media was prepared from the standard synthetic recipe described above, with the addition of mevalonate. Mevalonate was prepared via mevalonolactone saponification (Acros organics), by preparation of 1.5 mol KOH: 1 mol mevalonolactone dissolved in milliQ water and incubation at 37 °C for 2 h with constant rotation (Garcia, 2013; Martin et al, 2003). This solution was added to the media and the final pH was brought to 5.0 using 12 N HCl. A second media lacking mevalonate was prepared similarly; however, instead of the saponified mevalonate solution, 5 N KOH was added to mimic the mevalonate solution. This media was either used directly as a comparison to mevalonate-containing media or used to dilute the mevalonate media to achieve different concentrations of mevalonate. $OD_{600}$ and GFP were measured using a Tecan Infinite F Plex reader with the appropriate filter set ($OD_{600}$: 600 (10)), GFP: excitation 485(20), emission (535(25)). $OD_{600}$ measurements are reported using the unaltered units obtained from the Tecan plate reader, which are approximately tenfold lower than those measured with a spectrophotometer with a path length of 1 cm. To control for contamination, a blank media well was added to each plate for every growth experiment. The specific growth rate was determined from the linear portion of the logarithmic growth phase in permissive condition according to: $\mu = \ln(x(t)/x_0)/t$. For consistency, this same time window was used for permissive, pulsed, or nonpermissive growth rate calculations. For

characterizing the growth of the OptoREP lineages, the same growth rate calculation was used, with the addition of the lag growth phase calculated through the tangent method. This metric was calculated using the Microbial lag calculator, using a local regression of three time points (Smug et al, 2024).

## OrthoRep

The SAWy700 strain was subject to OptoRep passaging. The experimental conditions for passaging were the same as the growth assays, except for a stronger 2% light intensity (~100 µmol m$^{-2}$ s$^{-1}$). Eight biologically independent colonies were subject to semi-permissive passaging. Following the final semi-permissive passage, 100 µl of 1.0 $OD_{600}$ cells was plated on mevalonate-containing media under dark conditions. The p1 plasmid from colonies which maintained the parent light-dependent phenotype (by a light/dark growth challenge) was amplified using Phire Plant PCR mix, and ~3 colonies from each lineage and condition were subjected to Sanger sequencing. For Phire PCR amplification, overnight cultures were diluted 1:5 in MilliQ water and 2 µl was added to a 20 µl reaction containing 0.5 µM primer (SAWpri657). PCR cycling was performed according to the manufacturer's guidelines. Mutations which occurred at least once from each lineage are reported, with no other observable mutation within the *JEN1* reading frame.

## *JEN1* Y180 mutagenesis

Mutagenesis at the Y180 residue was performed via QuickChange PCR using primers containing a degenerate NNK codon. Specifically, the wild-type sequence was subjected to 25 cycles of PCR amplification, followed by DpnI digestion of the parent plasmid and direct transformation into the OptoMEV strain. Growth assays were performed as described above with slight modification: 100 µl culture volume in a 96-well plate, with a fixed 1:100 dilution rather than starting inoculation, grown for 24 h under dark conditions. The growth ratio for each colony was calculated for growth in mevalonate-supplemented media relative to growth in media lacking mevalonate under nonpermissive conditions, normalized to the same ratio for strains expressing the wild-type *JEN1*t transporter known not to recognize mevalonate. The growth ratio calculation is as follows: $y = (OD_{600+MVA}$library transformant/$OD_{600-MVA}$library transformant)/(average $OD_{600+MVA}$*JEN1*t/average $OD600-MVA$*JEN1*t). The CEN/ARS plasmid from strains which exhibited higher growth relative to the $JEN1^{Y180C}$ mutation was extracted using previously reported methods and reintroduced into the OptoMEV strain (Zhang et al, 2022).

## Fermentation, mevalonate conversion, and consortia

For fermentation, biologically independent colonies were grown overnight in 1 ml media and a 24-well plate at 30 °C and 200 RPM. Strains were then diluted 1:1 and inoculated at 0.1 starting $OD_{600}$ into 2 ml of media in a Greiner 12 ml cell culture tube. For strains with *ERG9* repression, outgrowth was performed in media lacking methionine. Cells were grown for 24 h, after which the cells were spun at 1.2 K RPM for 10 min, the media was aspirated, and cells were subjected to fermentation. Fermentation media for standard fermentations were SC-URA or SC-complete 2% glucose. For

mevalonate feeding fermentation, the media was as described for the growth assay, with the addition of a citrate-phosphate buffer and final pH of 6.0 (SC-URA, 2% glucose, 10 mM MVA or pH equivalent KOH, 0.5× CPB, pH 5). The 0.5× citrate-phosphate buffer was prepared as previously reported (Prins and Billerbeck, 2021). In strains lacking *HMG1/2*, 2 mM mevalonate was included in the overnight and outgrowth media to support growth, cells were washed in mevalonate-free media prior to fermentation. A 10% v/v dodecane overlay was added to fermentations, following which the cells were incubated for 48 h at 30 °C with 200 RPM shaking. Metabolite quantification was as described below (analytical methods) for all fermentations.

Consortia characterization and production fermentations were performed similarly with minor modifications. The media was modified to contain 2 mM added methionine, as per our previous mevalonate-secreting fermentations (SC-complete, 2%-glucose, 0.5× CPB, pH 6, 2 mM methionine (Wegner et al, 2021). For consortia fermentations, upstream and downstream strains were first outgrown overnight separately. The next day, cell densities from each overnight culture were measured and volumes from each culture were mixed to make different initial ratios (0, 10%, 20%, 30%, 40%, 50%, 60%, 70%, 80%, 90%, 100% upstream strain fractions) calculated for the total 2 ml fermentation volume. After both strains were mixed in 50 mL conical tubes, the tubes were spun down, resuspended in fresh media and incubated for 48-h fermentation as described above. Following fermentation, to estimate the final consortia compositions, 10 µl of cell co-culture was removed from the aqueous layer, diluted 1:100 twice, and 10 µl was plated for colony counting. SC-HIS-LEU-MET-TRP plates were used to select for the downstream strain, while SC-HIS-LEU-MET was used for total colony counts. Final consortia compositions were expressed as the fraction of upstream colonies (1-fraction downstream). Strains were incubated for 48 h at 30 °C prior to colony counting.

## Analytical methods

A volume of 0.7 ml of the fermentation broth was sampled for quantification of mevalonate. The broth was clarified via centrifugation (13.3 K RPM, 30 min) and frozen at −20 °C until analysis. Mevalonate was analyzed by HPLC (Agilent 1260 Infinity instrument; Agilent Technologies) with resolution by Aminex HPX-87H ion-exchange column (Bio-Rad). The column with eluted with 5 mM sulfuric acid at 55 °C with a flow rate of 0.6 ml/min. Analytes were quantified using a refractive index detector (RID), with concentrations calculated relative to peak areas of standard solutions.

Farnesene was quantified by extraction of the dodecane layer via solubilization with ethyl acetate. In all, 0.9 ml of ethyl acetate was added to the fermentation flask following aqueous metabolite extraction, and vortexed for 15 min prior to 1.2 K RPM centrifugation for 30 min. 0.5 ml of the upper organic layer was sampled for analysis using an Agilent 7890B GC system with a DB-Wax column (30 m length, 0.25 mm diameter, 0.5-µm film). Samples were subject to a split injection (0.5 µl, 1:20 split ratio) and resolved using constant 1.5 ml/min helium flow and the following oven gradient: Initial oven temperature was set to 70 °C and held for 3 min, and temperature was then ramped at a rate of 20 °C/min to 230 °C and held for 5 min. Quantification was performed using

flame-ionization detection (300 °C, H$_2$ flow 30 mL/min, airflow 400 mL/min, makeup flow 25 mL/min) relative to a bona fide standard kindly provided by Amyris Inc.

## Structural and data analysis

The predicted Jen1pt structure was obtained from AlphaFold DB (AlphaFold ID: AF-P36035) (Jumper et al, 2021; Varadi et al, 2022). Molecular docking of mevalonate (PDB: MEV) was performed on the predicted AlphaFold Jen1pt structure and the previously determined cryo-EM structure of lactate-bound MCT1 (PDB: 6lz0) (Wang et al, 2021). Jen1pt mutants were constructed in UCSF Chimera using the most probable predicted rotamer based on the Dunbrack library (Shapovalov and Dunbrack, 2011). Molecular docking was performed in PyRx using AutoDock Vina with a 25$^2$ - Å grid, with the residues of interest centered around the base of the grid (Dallakyan and Olson, 2015; Trott and Olson, 2010). Nine conformational isomers were generated for each *JEN1* variant. Further computational modeling was performed using the Rosetta macromolecular modeling suite (Leman et al, 2020). The Positioning Proteins in Membranes methodology (PPM 3.0) was used to calculate the orientation and location of Jen1pt in membranes modeled with fungal membrane lipid composition for automatic detection of membrane-spanning regions (Lomize et al, 2022). RosettaMP was used to minimize the energy of the AlphaFold Jen1pt structure within the membrane environment using the FastRelax protocol to generate 100 structures with the *franklin2019* score function (Alford et al, 2020). The lowest-energy protein structure was then used for mutational analysis. The RosettaMP ΔΔG protocol was used to model Jen1pt$^{Y180C}$ and Jen1pt$^{Y180G}$ single-point mutations. Minimized wild-type, Y180C, and Y180G structures were then used as initial poses for ligand binding simulations. Initial positioning of mevalonate within Jen1pt and mutants was determined based on structural superposition to lactate bound within MCT1. The RosettaLigand application via ROSIE was used to obtain an estimate of free energy of mevalonate and lactate binding across Jen1pt wild-type and mutants (DeLuca et al, 2015; Leman et al, 2020). The ligand and receptor were treated as flexible within a 5 Å radii of the initial ligand positioning. Ligand conformers were generated by the Biochemical Library (Kothiwale et al, 2015; Leman et al, 2020). Ligand position, orientation, and torsions as well as sidechain torsions were optimized using Monte Carlo sampling across 200 independent runs, and the top ten lowest-energy conformations were identified. All visualizations were prepared in PyMOL.

Data analysis was performed in R software with statistical analysis using the rstatix package. No sample blinding was performed. For single comparisons, a two-sided Mann–Whitney *U* test was performed. For multiple comparisons, statistical analysis was performed by the Kruskal–Wallis test with Dunn's post hoc comparisons. Graphs were generated in either Microsoft Excel or R using the ggplot2 package, while pathway figures were created using Biorender.

## Data availability

Computational structure models of mevalonate bound to wild-type *JEN1t*, *JEN1t*$^{Y180C}$, and *JEN1t*$^{Y180G}$ are available on Model Archive (modelarchive.org) with accession codes ma-39yos, ma-hnfca, and ma-dqvhp, respectively.

The source data of this paper are collected in the following database record: biostudies:S-SCDT-10_1038-S44320-025-00113-5.

## Peer review information

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

## Acknowledgements

The authors and their research are supported by the U.S. Department of Energy, Office of Biological and Environmental Research (JLA—Award Number DE-SC0022155), as well as by the NIGMS of the National Institutes of Health under grant number T32GM007388 (SAW), and the National Science Foundation Graduate Research Fellowship Program (GRFP) under DGE-2039656 (VJ). We thank Kevin Xu for providing the stabilized farnesene synthase with yeast enhanced GFP immediately upstream the Aa_FS_C synthase. The funders had no role in study design, data collection and analysis, decision to publish, or preparation of the manuscript. The content of this article is solely the responsibility of the authors and does not necessarily represent the official views of the National Institutes of Health.

## Author contributions

**Scott A Wegner**: Conceptualization; Data curation; Formal analysis; Validation; Investigation; Visualization; Methodology; Writing—original draft; Project administration; Writing—review and editing. **Virginia Jiang**: Data curation; Formal analysis; Validation; Investigation; Visualization; Methodology; Writing—original draft. **Jeremy D Cortez**: Methodology, Writing—review and editing. **José L Avalos**: Conceptualization; Formal analysis; Supervision; Funding acquisition; Investigation; Methodology; Writing—original draft; Project administration; Writing—review and editing.

Source data underlying figure panels in this paper may have individual authorship assigned. Where available, figure panel/source data authorship is listed in the following database record: biostudies:S-SCDT-10_1038-S44320-025-00113-5.

## Disclosure and competing interests statement

The authors have filed a patent for the OptoRep method and the mevalonate importers reported in this study.

# Expanded View Figures

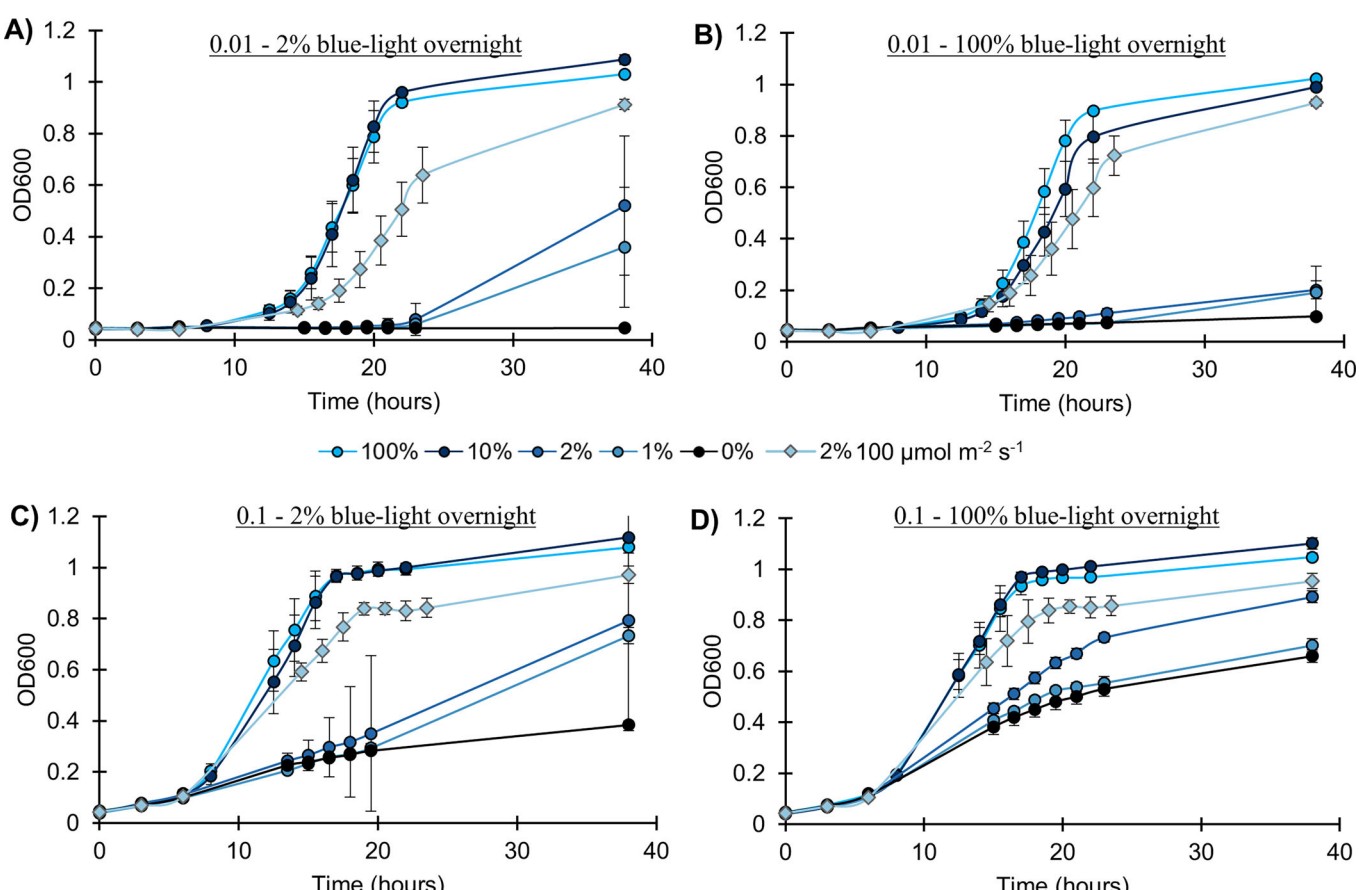

**Figure EV1. OptoMEV growth in different light doses and starting cell densities.**

(A, B) Cultures of the OptoRep-JEN1t strain (SAWy700) inoculated at an initial $OD_{600} = 0.01$ from overnight cultures grown in either (A) 2% light (2 s on/100 s total) or (B) 100% light, and then incubated to grow under different light conditions. (C, D) Cultures of the OptoRep-JEN1t strain (SAWy700) inoculated at an initial $OD_{600} = 0.1$ from overnight cultures grown in either (C) 2% light (2 s on/100 s total) or (D) 100% light, and then incubated to grow under different light conditions. Optical density measurements are reported in Tecan absorbance units (see "Methods"). Where not explicitly shown, light intensity varied between 40 and 60 µmol m$^{-2}$ s$^{-1}$. For all conditions light intensity in the overnight was equivalent to intensities in the experiment (i.e., between 40 and 60 µmol m$^{-2}$ s$^{-1}$ except for the 100 µmol m$^{-2}$ s$^{-1}$ condition). Average and individual data points for 4 biologically independent replicates are shown with standard deviation for each measured time point.

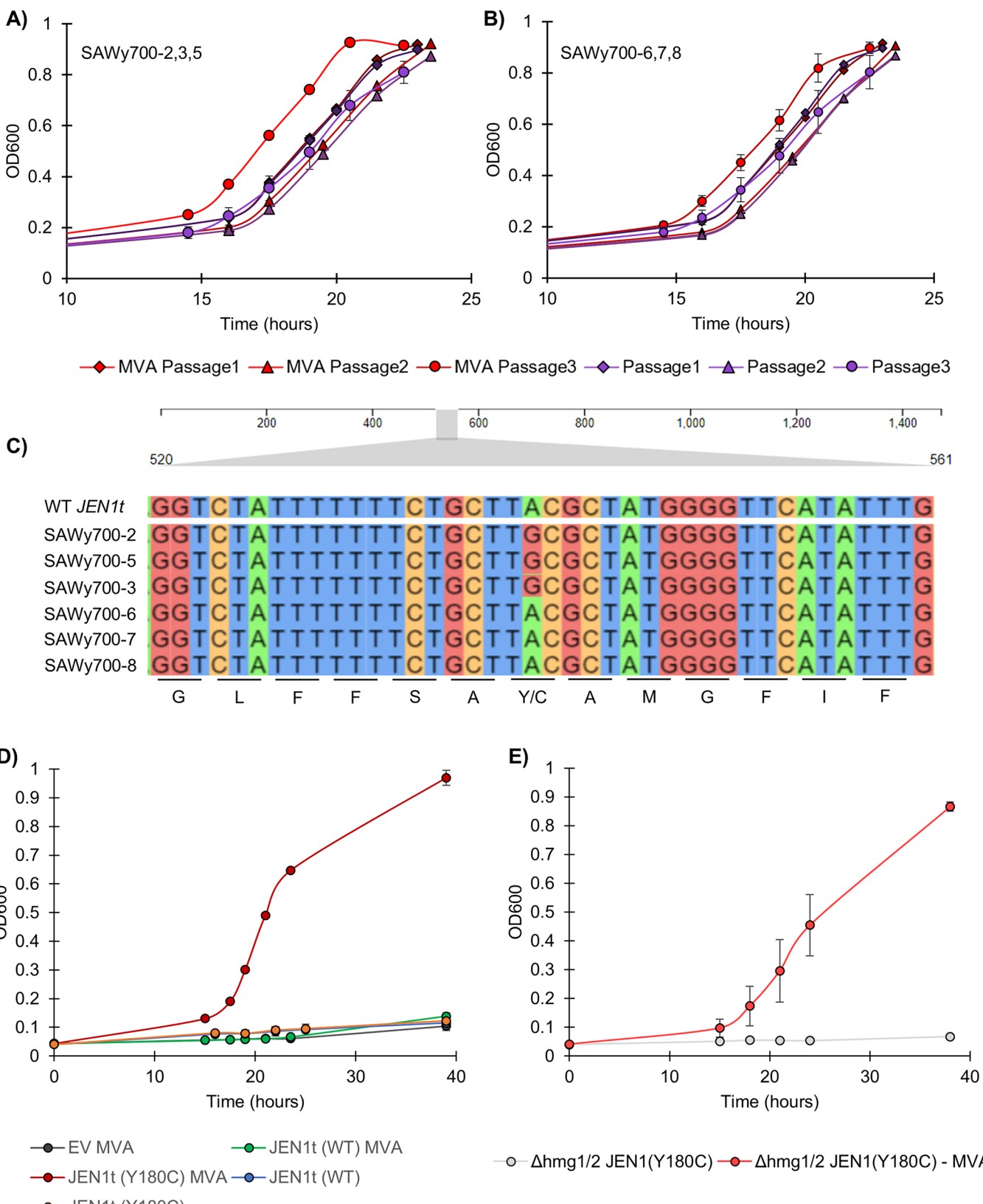

**Figure EV2.  OptoRep-*JEN1t* lineages with improved growth in mevalonate and *JEN1t* mutations obtained with OptoRep.**

(A) Averaged growth profiles of SAWy700-2, SAWy700-3, and SAWy700-5, from which *JEN1t*$^{Y180C}$ evolved independently three times, show a significant decrease in lag relative to (B) the averaged growth profiles of SAWy700-6, SAWy700-7, and SAWy700-8, from which no mutant sequence was identified. MVA refers to growth conditions supplemented with 10 mM mevalonate. Data points for (A, B) show the mean for the measured OD$_{600}$ of the three described lineages, with the standard deviation only the third passage shown (some error bars are smaller than the data point icons). (C) Individual colonies were isolated after purifying selection and their p1 plasmids sequenced, resulting in only one and the same transition mutation A539G (corresponding to A818 of wild-type full-length *JEN1*), equating in a Y180C mutation in the *JEN1t* amino acid sequence (corresponding to Y273 in the full-length *JEN1* protein). Mutations which occurred in at least one of the sequenced colonies are reported. (D) The *JEN1t*$^{Y180C}$ mutation was expressed in the OptoMEV strain using a CEN/ARS plasmid and assayed for mevalonate-dependent growth (MVA—10 mM mevalonate) in nonpermissive conditions (complete darkness). The growth curve was compared to a strain containing the wild-type *JEN1t* ((WT) MVA; SAWy711), an empty vector (EV MVA; SAWy705), or the strain expressing *JEN1t*$^{Y180C}$ (SAWy712) but incubated in medium without mevalonate. Conditions without MVA refer to unsupplemented medium. (E) The mevalonate-dependent growth of an HMG-CoA reductase-null genetic background (Δhmg1/Δhmg2), expressing the *JEN1t*$^{Y180C}$ transporter (SAWy715). Optical density measurements are reported in Tecan absorbance units (see "Methods"). The average for 4 biologically independent replicates are shown with standard deviation (some error bars are smaller than the data point icons) for each measured time point in (D, E).

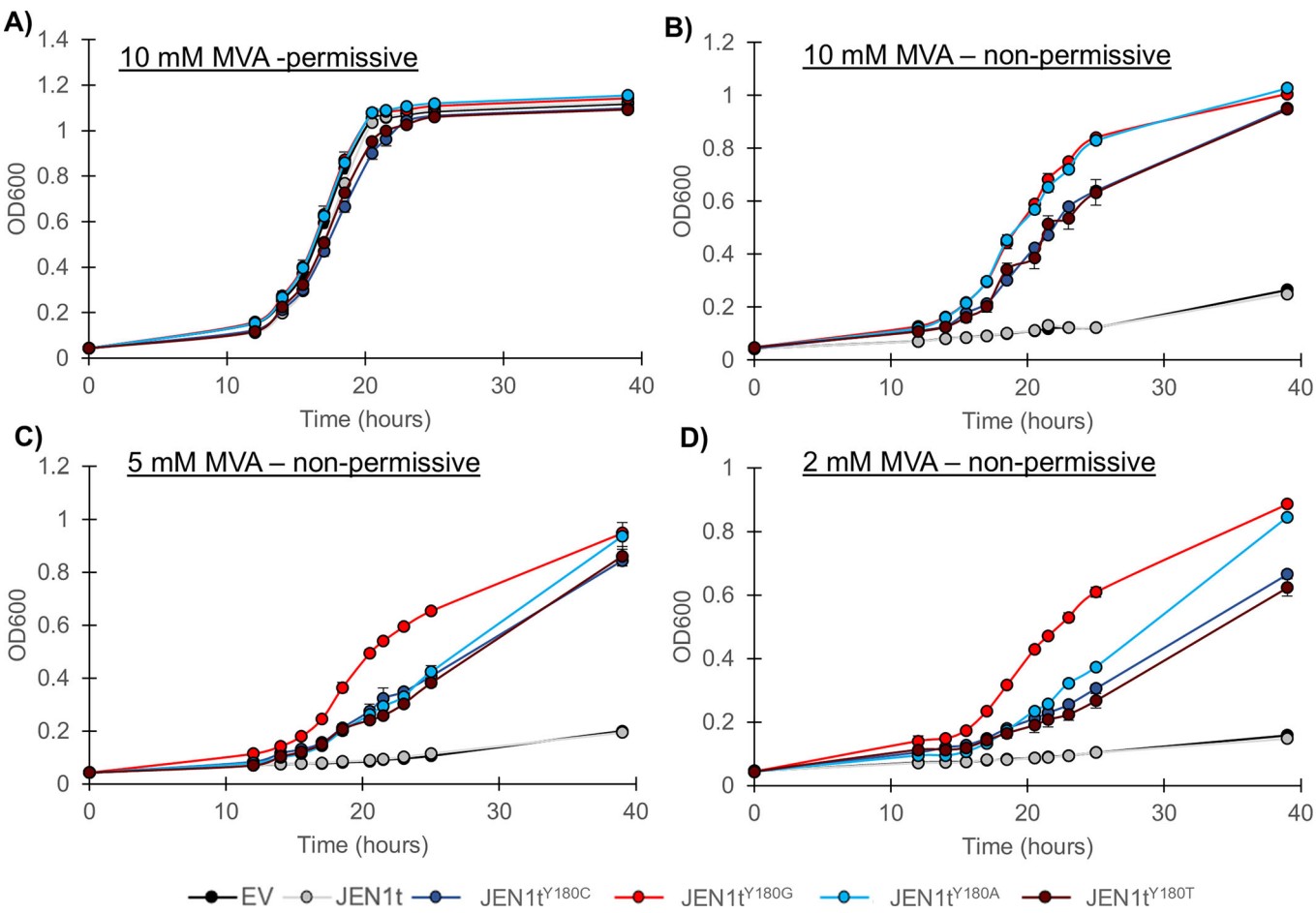

**Figure EV3.  Validation of Y180 NNK library hits.**

(A–D) The observed mutations plus controls were introduced into the OptoMEV strain (Empty Vector: SAWy705, $JEN1t^{Y180}$: SAWy711, $JEN1t^{Y180C}$: SAWy712, $JEN1t^{Y180G}$: SAWy738; $JEN1t^{Y180A}$: SAWy739, $JEN1t^{Y180T}$: SAWy740) and growth was assayed under different mevalonate (MVA) concentrations as listed. Permissive blue light intensity was ~40 µmol m$^{-2}$ s$^{-1}$. Optical density measurements are reported in Tecan absorbance units (see "Methods"). Average and individual data points for 4 biologically independent replicates are shown with standard deviation for each measured time point.

     

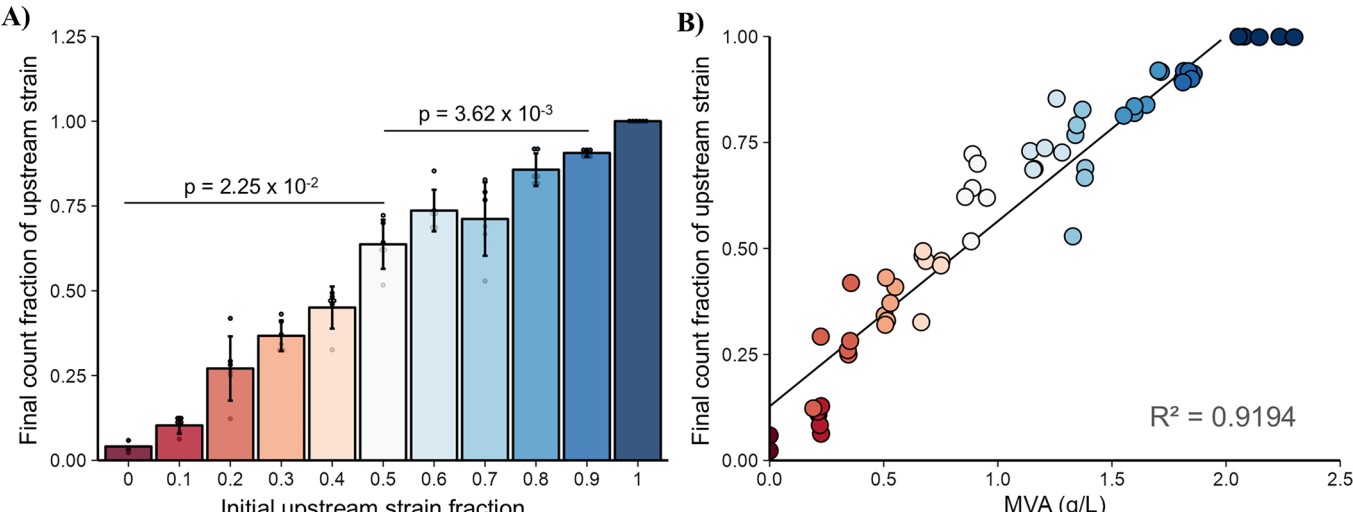

**Figure EV4. Final compositions of mevalonate-exchange consortia for farnesene production.**

(A) The final count fraction of the upstream strain (SAWy733) from consortia initiated with different initial fractions of the upstream strain. Serial dilutions for colony counting were performed post-fermentation with selection for either the downstream strain (SAWy749; SC-HIS-LEU-MET-TRP) or both consortia members combined (SC-HIS-LEU-MET). (B) correlation between the final colony count fraction of upstream strain and mevalonate (MVA) production by the consortia, with the Pearson coefficient of determination shown. Points are colored according to the initial upstream strain fraction of the consortia (same as A). Average and individual data points for 6 biologically independent replicates are shown with error bars showing the standard deviation. Data analysis was performed by Kruskal–Wallis with Dunn's post hoc comparisons for all panels.

