## [Peer Review File · Molecular Systems Biology]

Orthogonal replication with optogenetic selection evolves yeast JEN1 into a mevalonate transporter

Scott Wegner, Virginia Jiang, Jeremy Cortez, and Jose Avalos

Corresponding author: Jose Avalos (javalos@princeton.edu)

Review Timeline:

Submission Date:	3rd Aug 24
Editorial Decision:	26th Aug 24
Revision Received:	5th Mar 25
Editorial Decision:	27th Mar 25
Revision Received:	26th Apr 25
Accepted:	2nd May 25

Editor: Poonam Bheda

Transaction Report:

26th Aug 2024

Manuscript Number: MSB-2024-12530

Title: Orthogonal replication with optogenetic selection yields a JEN1 mutation for mevalonate transport.

Dear Dr. Avalos,

Thank you for the submission of your manuscript to Molecular Systems Biology. We have now received feedback from the three reviewers who agreed to evaluate your manuscript. As you will see from the reports below, the referees acknowledge the interest of the study and are overall supporting publication of your work pending appropriate revisions.

I think that the recommendations of the reviewers are rather clear and I therefore do not see the need to repeat the comments listed below. One of the more fundamental points raised by both Reviewers 1 and 3 is the relatively short timeframe of evolution of Jen1, where they suggest testing Jen1 that has been evolved for additional passages.

All other issues raised would need to be satisfactorily addressed. Please let me know in case you would like to discuss in further detail any of the comments, I would be happy to schedule a call.

We require:

1) A .docx formatted version of the manuscript text (including legends for main figures, EV figures and tables). Please make sure that the changes are highlighted to be clearly visible. Alternatively you may choose to submit your manuscript as a LaTeX file.

4) A .docx formatted letter INCLUDING the reviewers' reports and your detailed point-by-point responses to their comments. As part of the EMBO Press transparent editorial process, the point-by-point response is part of the Peer Review File (PRF), which will be published alongside your paper.

5) A complete author checklist, which you can download from our author guidelines (<https://www.embopress.org/page/journal/17574684/authorguide#submissionofrevisions>). Please insert information in the checklist that is also reflected in the manuscript. The completed author checklist will also be part of the PRF.

6) Please note that all corresponding authors are required to supply an ORCID ID for their name upon submission of a revised manuscript.

7) It is mandatory to include a 'Data Availability' section after the Materials and Methods. Before submitting your revision, primary datasets produced in this study need to be deposited in an appropriate public database, and the accession numbers and database listed under 'Data Availability'. Please remember to provide a reviewer password if the datasets are not yet public (see <https://www.embopress.org/page/journal/17574684/authorguide#dataavailability>).

In case you have no data that requires deposition in a public database, please state so in this section. Note that the Data Availability Section is restricted to new primary data that are part of this study. This study includes no data deposited in external repositories.

8) All Materials and Methods need to be described in the main text using our 'Structured Methods' format, which is required for all research articles. According to this format, the Methods section includes a Reagents and Tools Table (listing key reagents, experimental models, software and relevant equipment and including their sources and relevant identifiers) followed by a Methods and Protocols section describing the methods using a step-by-step protocol format. The aim is to facilitate adoption of the methodologies across labs. More information on how to adhere to this format as well as a downloadable template (.docx) for the Reagents and Tools Table can be found in our author guidelines: <https://www.embopress.org/page/journal/17444292/authorguide#structuredmethods>

An example of a Method paper with Structured Methods can be found here:
<https://www.embopress.org/doi/10.15252/msb.20178071>.

9) For data quantification: please specify the name of the statistical test used to generate error bars and P values, the number (n) of independent experiments (specify technical or biological replicates) underlying each data point and the test used to calculate p-values in each figure legend. The figure legends should contain a basic description of n, P and the test applied. Graphs must include a description of the bars and the error bars (s.d., s.e.m.). Please provide exact p values.

10) Our journal encourages inclusion of *data citations in the reference list* to directly cite datasets that were re-used and obtained from public databases. Data citations in the article text are distinct from normal bibliographical citations and should directly link to the database records from which the data can be accessed. In the main text, data citations are formatted as follows: "Data ref: Smith et al, 2001" or "Data ref: NCBI Sequence Read Archive PRJNA342805, 2017". In the Reference list, data citations must be labeled with "[DATASET]". A data reference must provide the database name, accession number/identifiers and a resolvable link to the landing page from which the data can be accessed at the end of the reference. Further instructions are available at .

11) We replaced Supplementary Information with Expanded View (EV) Figures and Tables that are collapsible/expandable online. A maximum of 5 EV Figures can be typeset. EV Figures should be cited as 'Figure EV1, Figure EV2' etc... in the text and their respective legends should be included in the main text after the legends of regular figures.

<https://www.embopress.org/page/journal/17574684/authorguide#expandedview>

13) Author contributions: CRediT has replaced the traditional author contributions section because it offers a systematic machine readable author contributions format that allows for more effective research assessment. Please remove the Authors Contributions from the manuscript and use the free text boxes beneath each contributing author's name in our system to add specific details on the author's contribution. More information is available in our guide to authors.

14) Disclosure statement and competing interests: We updated our journal's competing interests policy in January 2022 and request authors to consider both actual and perceived competing interests. Please review the policy

<https://www.embopress.org/competing-interests> and update your competing interests if necessary.

Please also suggest a striking image or visual abstract to illustrate your article as a PNG file 550 px wide x 300-600 px high. Share synopsis text and image, as well as eTOC:

Please note that these would be the final versions and changes during proofing are usually not allowed

16) As part of the EMBO Publications transparent editorial process initiative (see our policy here:

https://www.embopress.org/transparent-process#Review_Process), Molecular Systems Biology will publish online a Peer Review File (PRF) to accompany accepted manuscripts.

In the event of acceptance, this file will be published in conjunction with your paper and will include the anonymous referee reports, your point-by-point response and all pertinent correspondence relating to the manuscript. Let us know whether you agree with the publication of the PRF and as here, if you want to remove or not any figures from it prior to publication.

Please note that the Authors checklist will be published at the end of the PRF.

Molecular Systems Biology has a "scooping protection" policy, whereby similar findings that are published by others during review or revision are not a criterion for rejection. Should you decide to submit a revised version, I do ask that you get in touch after three months if you have not completed it, to update us on the status.

I look forward to receiving your revised manuscript.

Yours sincerely,

Poonam Bheda, PhD
Scientific Editor
Molecular Systems Biology

Reviewer #1:

Summary:

In vivo mutagenesis allows to access new phenotypes that cannot be attained through traditional adaptive laboratory evolution techniques. A major caveat in adaptive laboratory evolution experiments is to set up an adequate selective pressure. This can be particularly challenging for systems with low basal activity and poor growth performance. In this study, Wegner et al. alleviate this problem by combining the in vivo mutagenesis tool OrthoRep with the EL222 optogenetics platform to fine-tune the selection pressure while maintaining adequate growth for selection. They show the utility of their tool to evolve a truncated version of the native *S. cerevisiae* JEN1t transporter (JEN1tY180C) to allow import of mevalonate, a key metabolite relevant for many biotechnological applications. Guided by the localisation of the mutation close to the substrate binding pocket, they further used a saturation library at this location to yield mutants with superior mevalonate uptake and characterize their properties at structural level. Finally, the utility of these improved mevalonate importers is demonstrated during mono- and co-culture farnesene production.

General remarks:

In vivo mutagenesis has gained traction in the last couple of years as a powerful platform to evolve new phenotypes. Combining this with optogenetics represents a conceptual advance, allowing for more precise and dynamic fine-tuning than other systems can afford (f.i. dCas9 repression). In this sense, this study is novel and timely for the metabolic engineering community. Yet, some parts of the manuscript are lacking clarity and adequate controls.

First, uptake of mevalonate through the evolved Jen1pt is never directly shown but rather inferred. This is problematic since validation is not performed in the adequate background (independently from optoMEV, in absence of native Jen1, etc..). Importantly, presence of background activities involved in mevalonate synthesis or import are not fully accounted for and additional controls should be included.

Second, a critical aspect overlooked in this study is the stability of the system. Typical in vivo mutagenesis experiments last days (for instance 40 day with standard OrthoRep; 10.1038/s41467-020-19539-6). In this study, only a small number of passages were tested (3 passages, 24 hr each), resulting in fixation of only a handful of mutations (against ≈ 10 -15 for standard OrthoRep; 10.1038/s41467-020-19539-6). To prove the generalizability of this tool to other systems which would require longer evolving time, stability (i.e. stability of pC120-HMG1 expression) should be assessed for a higher number of passages.

Major points:

I.160 Absence of Jen1t can still lead to basal growth in the mevalonate-replete, dark condition as reported by the authors. However, no explanation is given despite the relevance of potential promiscuous activities of mevalonate metabolism later in the manuscript (either through mevalonate import or synthesis). Since some level of promoter leakiness is already observed in the dark, mevalonate-deplete condition (Fig. 1D), could the improved growth of optoMEV in Fig.1F be the result of residual or leaky Hmg1 expression? From the Methods section, it seems that all seed cultures are with light to allow for biomass accumulation. While exposure to darkness stops Hmg1 synthesis, existing Hmg1 remains and needs to be further diluted. Pre-culture of cells in dark conditions for a couple of generations (until growth-arrest) to account for Hmg1 dilution before the growth assays would be informative. Alternatively, if this residual growth is the result of background activities (such as unreported reaction promiscuity), can a $\Delta hmg1/\Delta hmg2$ strain grow in the presence of mevalonate? This is briefly mentioned I.184 but it is not clear if this was tested here or elsewhere.

I.693 Related to the above point, to add mevalonate to the media, the authors prepare a mevalonate solution based on saponification of mevalonolactone. The efficiency of this conversion under these conditions (2h at 37°C) is not reported. What is the conversion efficiency and could residual mevalonolactone impact downstream growth assays or even lead

to batch effects (i.e. when prepared on different days)?

I.193 Is the HMG1 expression plot (Fig. 2A, on the right) an actual experiment? If so, details of how HMG1 expression was measured is missing. Also, it seems there is a typo on the x axis or missing light conditions?

I.196 Please correct Fig. 2B. In its current form, the legend does not match the figure points/lines. For instance, "100" condition colouring is mixed with "2" condition. Also, it is not explained what these numbers (100,10,2,1) are. For instance, what is the difference between 2 and 2% 100 μM m^{-2} s^{-1} ? I couldn't find the explanation anywhere in text.

I.226 Growth improvements are displayed as a difference of OD (Fig. 2D). This should be complemented with a growth rate difference plot. This is the better and more usual metric to represent improved growth from adaptive laboratory experiments (f.i. as previously shown for OrthoRep 10.1038/s41467-020-19539-6) and less arbitrary than selecting an OD difference at a specific timepoint. Also, all replicates/lineages should be displayed to have a sense of the improvement variability that OrthRep can generate - which is also an interesting finding. Alternatively, explanation for replicate occlusion should be included in the figure legend.

I.237 From the Methods, 3 colonies were Sanger sequenced per lineage. Was the mutation present in all colonies or was there some heterogeneity? I would assume that some of these mutations not fully fixed in the population, even after 3 passages.

I.246 "To test whether the JEN1tY180C mutation is responsible for the increased growth under non-permissive light conditions with mevalonate complementation, we expressed it in the optoMEV strain using a CEN/ARS plasmid. The resulting strain JEN1tY180C grows robustly (at rate of $0.203 \pm 0.004 \text{ hr}^{-1}$) in non-permissive dark conditions in mevalonate supplemented medium, but not in media without mevalonate (Table S4, Fig S4D)." This is tested in a strain background containing optoMEV (SAWy524). To confirm the effectiveness of JEN1tY180C in enhancing mevalonate import, it should also be tested independently of the optoMEV system already at this stage. Also is this performed in a background with the native Jen1pt? Ideally, JEN1tY180C ability to rescue growth should be tested in a clean $\Delta\text{hmg1}/\Delta\text{hmg2}$ without native JEN1. Again, this would also clarify if mevalonate can be imported/synthesised by other means.

I.260 Maybe the author can clarify the advantages of using a saturation mutagenesis library (267 mutants) for position 180 given that testing 19 other strains (each with a different amino acid at this position) would be sufficient already.

I.277 Fig. 3B. This data is discrete (different MVA concentration, light/dark) and should be represented through bar plots and not as connecting lines. Fig. 3d Legend for losanges and circles should be displayed on the graph for better readability.

I.380 Please add missing significance comparisons when relevant and indicate as n.s. when not statistically significant. In Fig. 4D, can the author also clarify what is $\delta::\text{aaFS}$ and why is it shown here?

I.380 Fig. 4 should be complemented with additional data regarding the fermentation process. Fermentation time, starting OD, growth curve etc. It is difficult to judge from these bar plots the actual contribution of JEN1t. Panel B/D: I am confused by the strain genotype in this figure. Is the native JEN1t present there? If this is the case, then what's the contribution of the native vs mutated Jen1pt in farnesene increase?

I.431 The microbial consortium presented here goes against the cooperation and complementation principles associated with microbial consortia such as division of labour, where the production pathway is typically split between members. Here, both strains are able to produce the full farnesene pathway which limits the relevance of using such a consortium. This translates into a moderate (yet significant) increase in farnesene production. I am not convinced that this increase in farnesene titers is due to mevalonate exchange for two reasons: 1. In the absence of growth curves, this increase could be related to the nature of the growth competition between different ratio of upstream/downstream strains. 2. Mevalonate exchange between members of the consortia is inferred but not proven. By which mechanisms would mevalonate be exported from the upstream strain. It is represented by a simple arrow in Fig. 5B without further information. Is it through the native JEN1t?

Minor points:

I.186 "we introduced a glucose-insensitive mutant of JEN1". The authors could clarify the reasoning for selecting a glucose-insensitive mutant of JEN1. I.e. Presence of glucose leads to Jen1 endocytosis.

Supp I.52 / Fig. S3. Please change p1-p3 passage naming to something else to avoid confusion with p1 and p2 plasmids from OrthoRep. Also, meaning of "MVA" is missing. I suppose this represents 10mM mevalonate?

Supp Table S6. Shouldn't yEZ118 be SAWy118 in the strain description of SAWy119. Double check typos in strain description.

I.314/316s typos: This should read JEN1t. Make sure this is corrected throughout the text

I.455 Fig. 5A and 5B Nomenclature is not consistent between subpanels. Is it upstream mevalonate pathway or upper

mevalonate pathway?

Reviewer #2:

This is overall a strong study that adds both a general new tool/approach for continuous evolution (i.e., optoRep with optogenetic control of selection strength) and makes a notable engineering discovery of a new transporter for mevalonate based on JEN1. The work is very well written with clear logic. The work is also complete both in characterization of evolved transporters and their application in farnesene production, especially the microbial consortia format which is a smart and well-motivated variation. Experiments are complete and appear rigorous. Overall, I believe the work described and level of analysis carried out would be interesting to and enjoyed by MSB readers.

I have some minor issues to raise to guide revision...

1) I am not sure "gain-of-function" is the best term for what the authors mean. The authors are highlighting a case where a function is gained in a gene that did not have that function before (i.e., going from "zero to something"). But I believe gain-of-function is also commonly used to include cases where function is enhanced rather than emerging from none (i.e., going from "something to something more"). The authors rightfully wish to emphasize the fact that in their experiment, JEN1 had no detectable activity for mevalonate before but found a mutation that now gives it activity, which is indeed a unique, interesting, and noteworthy outcome. The fact that JEN1 has no activity also motivates the need for an ontogenetic control strategy to titrate down the function of another gene to drive selection. Perhaps the better term is neofunctionalization? I'm sure there is some vagueness in how gain-of-function is used, given it is also part of colloquial language. At the very least, I recommend the authors consider this point. If they agree that gain-of-function is not the optimal term, the authors should look at the few instances in which it is used in their manuscript and modify accordingly.

2) It would be helpful to explain why the MCT1 point mutant transporter that works in mammalian cells was not used here as the starting point for evolution, given it has some low activity in yeast. Also, it is interesting that we are also looking at a single aromatic to cysteine mutation for MCT1. Is that a coincidence or is it a structurally homologous position?

3) I understand why HMG2 was deleted. But I believe it should be made more clear both textually and in Figure 1. On first glance, I was confused why Figure 1D showed good growth in the dark for the Δ hmg2 strain. After some thinking, I figured out that the Δ hmg2 growth curves were simply the more matched control for optoMEV strains since those need to have Δ hmg2 deleted as only hmg1 was put under optocontrol but hmg2 is sufficient to make mevalonate even when hmg1 is not expressed. However, I think this can be made immediately clear to the reader by adding in the main text and figure legend. And by expanding the strain cartoons in 1A and B.

4) It would be good to have more discussion and information about the dependence of the [relationship between light exposure and selection strength] on [cell culture density].

5) I thought it was smart that the authors mutated Y180 after finding Y180C from OrthoRep in order to sample other amino acids at that position that are not accessed due to the biased nature of OrthoRep's TP-DNAP1-4-2. This resulted in the discovery of the Y180G mutant that was ultimately superior. It might be helpful for the authors to note that OrthoRep systems with higher transversion rates are available (Rix et al., 2023, which the authors cite) to make the Y to C to G path more likely. It might also have been the case that decreasing mevalonate after going to dark in the evolution experiment would have found C to G even though it is a less common mutation due to TP-DNAP1-4-2's bias.

7) I appreciate the docking experiments. But it should be noted that they are using an alphafold structure. The extent to which this creates liabilities remains to be worked out in the field. I would therefore recommend that the authors comment that readers should keep this in mind, even though what is docked seems sensible.

8) The authors should cite and give a brief note to this paper on transporter evolution when discussing the context of their work: <https://elifesciences.org/reviewed-preprints/93971>

9) One reality of the study is that in retrospect, the transporter single mutation could have likely been found by just using an error-prone PCR library on JEN1 and selecting for tolerance of HMG1/2 knockout in a population of JEN1 mutants plus mevalonate. Such an approach would have required neither OrthoRep nor optogenetic titration of the selection. Of course, this is only a realization in retrospect. But I think it would be helpful to mention this both as a nuance for readers to appreciate and a springboard to a further discussion of what types of adaptation or neofunctionalization would only be accessible with the optorep approach - ones requiring complex multi-mutational pathways or more complex selection histories involving fluctuating pressures, etc.

Reviewer #3:

Summary

Wegner et al. apply a combined approach of the continuous directed evolution system OrthoRep, optogenetics (OptoRep) and classical directed mutation to evolve a mevalonate importer from a truncated version of the monocarboxylate transporter JEN1. No mevalonate importer has yet been identified in yeast and a mammalian mutant transporter has only low activity in yeast. By evolution of a mevalonate importer for yeast, the authors prove the power of OptoRep. The evolved transporter is then used to support isoprene production in yeast (consortia).

General remarks

The manuscript is well written and understandable. The authors outline the engineering strategy well which makes it easy to follow it. In addition, the data presentation is comprehensive and in a reasonable format. The main value of this manuscript might be the application of optogenetics in OrthoRep to finetune expression of metabolites which (yet) cannot be important into cells. This allows to evolve transport activities which have not been engineered yet. The authors demonstrate and present this concept very well. It adds a useful feature to the OrthoRep tool kit. From an industrial point of view, evolving a mevalonate transporter might be of limited usefulness, however, it is a good proof-of-concept target to demonstrate the power of the newly developed OptoRep approach.

Major points

1. Using OrthoRep, the authors performed continuous directed evolution of JEN1 for three passages. This is a very short time to evolve a gene. As far as I can see, the evolved strain does not perform as well as the wildtype (around 70%), and it might be reasonable to perform the evolution for a couple more passages. Evolving the improved mutants (Y180G or Y180A) further in OrthoRep might have provided another strategy to improve the importer. Have the authors considered further improvement of the importer?
2. Did the authors compare the activity of the evolved transporter with the mutant form of the mammalian transporter, e.g. in growth assays? Is the evolved transporter more or less active than existing one(s)?

Minor points

1. I am surprised that the yeast strains level off in growth at OD 1. Is the optoMEV strain restrained in growth due to the optogenetic engineering?
2. Figure 1D shows that the optoMEV strain can grow somewhat in the dark (no MEV supplementation). Given a starting OD of 0.01 (?) it made four doublings. Is this a result of MEV carryover from the population from which it was inoculated or the TF EL222 leaky?

Response to reviewers

Dear Dr. Bheda,

Thank you for the opportunity to submit a revised draft of our manuscript "Orthogonal replication with optogenetic selection evolves *JEN1* into a mevalonate transporter" for publication in *Molecular Systems Biology*. We also thank the reviewers for their comments and suggestions, which have helped us improve the quality and impact of our study. Below, we provide our point-by-point responses to the reviewers' comments (in blue). **Please note that the line numbers referenced below correspond to the revised manuscript with tracked changes.** We believe the recommended modifications have enhanced the scientific merit of the manuscript.

Reviewer #1:

Summary:

In vivo mutagenesis allows to access new phenotypes that cannot be attained through traditional adaptive laboratory evolution techniques. A major caveat in adaptive laboratory evolution experiments is to set up an adequate selective pressure. This can be particularly challenging for systems with low basal activity and poor growth performance. In this study, Wegner et al. alleviate this problem by combining the in vivo mutagenesis tool OrthoRep with the EL222 optogenetics platform to fine-tune the selection pressure while maintaining adequate growth for selection. They show the utility of their tool to evolve a truncated version of the native *S. cerevisiae* JEN1t transporter (JEN1tY180C) to allow import of mevalonate, a key metabolite relevant for many biotechnological applications. Guided by the localisation of the mutation close to the substrate binding pocket, they further used a saturation library at this location to yield mutants with superior mevalonate uptake and characterize their properties at structural level. Finally, the utility of these improved mevalonate importers is demonstrated during mono- and co-culture farnesene production.

General remarks:

In vivo mutagenesis has gained traction in the last couple of years as a powerful platform to evolve new phenotypes. Combining this with optogenetics represents a conceptual advance, allowing for more precise and dynamic fine-tuning than other systems can afford (f.i. dCas9 repression). In this sense, this study is novel and timely for the metabolic engineering community. Yet, some parts of the manuscript are lacking clarity and adequate controls.

We thank the reviewer for the supportive comments.

First, uptake of mevalonate through the evolved Jen1pt is never directly shown but rather inferred. This is problematic since validation is not performed in the adequate background (independently from optoMEV, in absence of native Jen1, etc..). Importantly, presence of background activities involved in mevalonate synthesis or import are not fully accounted for and additional controls should be included.

Line numbers refer to the manuscript with tracked changes.

To show a more direct mevalonate uptake by the evolved Jen1pt, we now show mevalonate-dependent growth in a *hgm1Δ/hgm2Δ* strain independently of OptoMEV, as the reviewer suggested (**Lines 244-248** and Figure EV2E). Moreover, it is very well known that the native Jen1p is completely inactivated in glucose by glucose-induced protein endocytosis and degradation; therefore, our experiments (all done in glucose) are equivalent to the experiments in strains lacking the native Jen1p that the reviewer suggests (please see DOI: 10.1074/jbc.RA117.001062 and the studies cited therein).

Our experiments (**Lines 244-251**) show that JEN1t^{Y180C} expression and mevalonate feeding are both essential for growth of a strain lacking all endogenous HMG-CoA reductase activity (double deletion of *HGM1* and *HGM2*; Figure EV2E). Consistent with previous studies (doi: 10.1073/pnas.83.15.5563), we found that strains with the double deletion *hgm1Δ/hgm2Δ* are inviable and thus cannot be constructed for additional experiments for background measurements. This is due to the essential nature of the HMG-CoA reductase activity of the enzymes encoded by these genes, well-known to be required to produce the essential metabolite, mevalonate. Growth of this double deletion was shown to need complementation with a heterologous HMG-CoA reductase (doi: 10.1111/j.1365-2672.2008.04060.x.), reminiscent of (and functionally equivalent to) the complementation with JEN1t^{Y180C} plus mevalonate feeding needed in our study. Therefore, the fact that this strain is able to be constructed only when co-expressing JEN1t^{Y180C} and feeding mevalonate, is strong evidence that JEN1t^{Y180C} mediates mevalonate import. Nevertheless, to address the concern raised by the reviewer, we now note in the revised manuscript that the *hgm1Δ/hgm2Δ* strain could not be constructed because of the reasons mentioned above (**Lines 244-251**).

To address the concern about a missing a control showing the background import activity, we conducted new experiments to measure the contribution of wild-type *JEN1t* to cell growth through mevalonate complementation. These experiments show that overexpression of the truncated glucose-stabilized form of *JEN1t* (wildtype - without mutations that recognize mevalonate) is equivalent to the growth defect observed in the empty vector control under non-permissive conditions (Figure 3B). This is further supported by additional experiments (Figure EV2D) where overexpression of *JEN1t* (wildtype) shows no difference in growth between mevalonate supplemented versus not supplemented non-permissive conditions and is equivalent to empty vector.

Further demonstration of Jen1pt mutants “directly” uptaking mevalonate would be far from trivial. Perhaps we misunderstood the reviewer’s comment but determining mevalonate uptake biochemically would require the overexpression, purification, and functional reconstitution of Jen1t and at least one of its mutants. Membrane proteins, such as Jen1t and its mutants, are notoriously difficult to overexpress and purify at levels required for biochemical experiments, and their reconstitution would need to be done in lipid vesicles, which would need to be optimized. This biochemical characterization would add significant delay to the publication of this study.

In any case, genetic evidence for metabolite uptake by engineered transporters, similar to the one we present here, has been widely accepted for decades. For example, for engineered xylose importation (<https://doi.org/10.1186/s13068-014-0168-9>, <https://doi.org/10.1186/s12934-020-01354-9>, <https://doi.org/10.1073/pnas.1311970111>), cellobiose importation (<https://doi.org/10.1073/pnas.1010456108>), or quinic acid importation (<https://doi.org/10.1021/acssynbio.1c00229>).

Second, a critical aspect overlooked in this study is the stability of the system. Typical *in vivo* mutagenesis experiments last days (for instance 40 day with standard OrthRep; 10.1038/s41467-020-19539-6). In this study, only a small number of passages were tested (3 passages, 24 hr each), resulting in fixation of only a handful of mutations (against ≈ 10 -15 for standard OrthoRep; 10.1038/s41467-020-19539-6). To prove the generalizability of this tool to other systems which would require longer evolving time, stability (i.e. stability of pC120-HMG1 expression) should be assessed for a higher number of passages.

The reviewer raises an interesting question. It should be noted, however, that it is quite remarkable that in 3 independent evolutionary campaigns, the same Y180C emerged as the sole mutation present and sufficient to enable growth in non-permissive conditions with mevalonate supplementation. The fact that none of our campaigns took as long as what has been required in other studies to evolve complementary mutations in their targets of interest is a testament to the importance of residue Y180 for substrate recognition, the accessibility of enabling mutations at that position of the gene, and the selection pressure that our optogenetic system is able to exert. We believe that artificially extending the evolutionary campaigns by conducting additional passages (longer than needed to obtain a gain-of-function mutation that confers a sufficiently robust growth phenotype in non-permissive conditions) would not be very informative and only add significant delay in publication. Such experiment would not change the main conclusions of the study: the role of Y180 in substrate recognition in Jen1p and the ability of OptoOrthoRep to regulate the selective pressure in difficult-to-evolve systems in order to obtain gain-of-function mutations. It would also not add any new insights into the system, including on the stability of pC120-HMG1 expression, as the reviewer suggests. While additional passages might lead to additional mutations in *JEN1t*, most of them would probably not confer additional benefits to mevalonate import. (In this regard, we view the repeated emergence of only a single mutation to confer mevalonate importation activity on Y180 as a major advantage because it was not necessary to deconvolute the individual and synergistic contributions of multiple mutations). Moreover, and directly to the reviewer's point, as soon as the Y180C mutation emerges, the selection pressure to maintain (or lose) the pC120-HMG1 construct immediately changes, so extended campaigns with *JEN1t* would not measure the true stability of this construct. However, based on previously published studies, we have a sense that EL222/pC120 can stably control essential genes (e.g. *PDC1*) continuously for at least 11 days in fed-batch fermentations, or about ~ 180 generations (<https://doi.org/10.1038/nature26141>).

Nevertheless, to address the reviewer's concern, we have added in the discussion new text disclosing the limitation that the stability of pC120-controlled constructs in longer evolutionary campaigns remains to be determined: "one potential limitation of this study is that, because the

Y180C mutation consistently emerged relatively quickly compared to other OrthoRep studies, we were unable to measure the stability of the pC120-*HMGI* construct in evolutionary campaigns longer than the three passages (or ~72h) that it took for this mutation to emerge. Therefore, the stability of pC120-controlled constructs in projects that need additional passages (e.g., that require the accumulation of multiple mutations) remains to be determined. While in previous studies we have observed a relatively long stability of essential genes (such as *PDC1* in glucose) controlled by pC120 or derived circuits, this stability is liable to vary depending on specific features of the system and the light-conditions used to tune the selective-pressure on the evolutionary campaign.” (Lines 456-465)

Major points:

l.160 Absence of *Jen1t* can still lead to basal growth in the mevalonate-replete, dark condition as reported by the authors. However, no explanation is given despite the relevance of potential promiscuous activities of mevalonate metabolism later in the manuscript (either through mevalonate import or synthesis). Since some level of promoter leakiness is already observed in the dark, mevalonate-deplete condition (Fig. 1D), could the improved growth of optoMEV in Fig.1F be the result of residual or leaky Hmg1 expression? From the Methods section, it seems that all seed cultures are with light to allow for biomass accumulation. While exposure to darkness stops Hmg1 synthesis, existing Hmg1 remains and needs to be further diluted. Pre-culture of cells in dark conditions for a couple of generations (until growth-arrest) to account for Hmg1 dilution before the growth assays would be informative. Alternatively, if this residual growth is the result of background activities (such as unreported reaction promiscuity), can a $\Delta hmg1/\Delta hmg2$ strain grow in the presence of mevalonate? This is briefly mentioned l.184 but it is not clear if this was tested here or elsewhere.

The reviewer is correct. The basal growth of OptoMEV in the dark observed in Figures 1D and 1F could be due to leaky expression of *HMGI* from the P_{C120} promoter. To address this concern, we have added this explanation to the manuscript (Lines 171-172 and 1062-1063).

The reviewer is also correct about the need to dilute Hmg1p in order to obtain consistent differences in growth between permissive and nonpermissive conditions. Rather than pre-culturing cells, however, we used high levels of back-dilution (low starting inoculums of 0.01) such that residual Hmg1p is reduced through mitotic dilution. This is now more clearly explained in the revised manuscript (Line 153-154).

l.693 Related to the above point, to add mevalonate to the media, the authors prepare a mevalonate solution based on saponification of mevalonolactone. The efficiency of this conversion under these conditions (2h at 37C) is not reported. What is the conversion efficiency and could residual mevalonolactone impact downstream growth assays or even lead to batch effects (i.e. when prepared on different days)?

Mevalonolactone saponification with KOH is a well-established and widely used procedure in organic chemistry. It is highly efficient and typically assumed to reach completion. The study that reported the protocol we used (**line 671-674**), which uses a 2-hour reaction time, confirmed mevalonolactone conversion by TLC. This protocol is in fact derived from an earlier protocol that used only 30 minute reactions (Campos N. *Biochem. J.* (2001) 353, 59-67:PMID:11115399). Other studies have also successfully used only 30-minute reactions (<https://doi.org/10.1186/s12934-016-0622-4>). Therefore, by using this well-established method with a prolonged reaction time of 2 hours, as reported in (<https://doi.org/10.1038/nbt833>), we have no concern of incomplete conversion, batch-to-batch variation, or impact on growth assays.

1.193 Is the HMG1 expression plot (Fig. 2A, on the right) an actual experiment? If so, details of how HMG1 expression was measured is missing. Also, it seems there is a typo on the x axis or missing light conditions?

We regret that this figure caused confusion and has now been removed. The figure was simply a schematic of how we hypothesize the system works based on previous studies as well as the growth observed in the new Fig 2A. We believe the new figure layout provides the reader with more meaningful and less confusing information.

1.196 Please correct Fig. 2B. In its current form, the legend does not match the figure points/lines. For instance, "100" condition colouring is mixed with "2" condition. Also, it is not explained what these numbers (100,10,2,1) are. For instance, what is the difference between 2 and 2% 100 $\mu\text{mol m}^{-2} \text{s}^{-1}$? I couldn't find the explanation anywhere in text.

Thank you for catching this error. The coloring has been corrected. The numbers 100, 10, 2, 1, refer to the light dose as the percent of light being on over a specified forcing period; for example, 10% over a 100 s forcing period would be 10 seconds on followed by 90 seconds off, repeated over the course of the experiment (a light duty cycle of 10%). The difference between 2% and 2% 100 $\mu\text{mol m}^{-2} \text{s}^{-1}$ marked on the figure refers to the difference in light intensity in those light pulses. While both experiments used a light duty cycle of 2% over 100 s forcing period (2s on/ 98s off), the first one used a lower light intensity of 40-60 $\mu\text{mol m}^{-2} \text{s}^{-1}$, compared to the first one that used 100 $\mu\text{mol m}^{-2} \text{s}^{-1}$. This, light dose can be varied by both different duty cycles (the frequency of light pulses over a specified forcing period), and the light intensity of those light pulses, both of which were exploited to achieve an intermediate growth curve in these plots. We have added further details to the figure legend to help clarify these points (**Lines 1070-1074**).

1.226 Growth improvements are displayed as a difference of OD (Fig. 2D). This should be complemented with a growth rate difference plot. This is the better and more usual metric to represent improved growth from adaptive laboratory experiments (f.i. as previously shown for OrthoRep 10.1038/s41467-020-19539-6) and less arbitrary than selecting an OD difference at a specific timepoint. Also, all replicates/lineages should be displayed to have a sense of the

improvement variability that OrthRep can generate - which is also an interesting finding. Alternatively, explanation for replicate occlusion should be included in the figure legend.

We agree that the original figure was not sufficiently informative. The revised manuscript now provides a growth rate difference plot, as requested by the reviewer, averaged for all six lineages (Fig 2E), as well as lag time difference plot (Fig 2D), which shows a more significant change across passages. We attribute the relatively small increase in growth rate between passages to incomplete penetrance of the mutation in the passage population, as further discussed in the next comment.

In addition, the growth curves for each individual lineage are shown in Appendix Figure 2 and the averaged growth curves for all passages of the successful lineages SAWy700-2,3,5,6,7,8 are shown in Figure 2C.

1.237 From the Methods, 3 colonies were Sanger sequenced per lineage. Was the mutation present in all colonies or was there some heterogeneity? I would assume that some of these mutations not fully fixed in the population, even after 3 passages.

The reviewer is indeed correct. In the SAWy700-2,3,5 lineages at least one colony exhibited the Y180C mutation but not all colonies did, while all colonies analyzed from the other lineages returned wild-type sequence. This has now been specified (**Line 226–229**).

1.246 "To test whether the JEN1tY180C mutation is responsible for the increased growth under non-permissive light conditions with mevalonate complementation, we expressed it in the optoMEV strain using a CEN/ARS plasmid. The resulting strain JEN1tY180C grows robustly (at rate of 0.203 +/- 0.004 hr⁻¹) in non-permissive dark conditions in mevalonate supplemented medium, but not in media without mevalonate (Table S4, Fig S4D)." This is tested in a strain background containing optoMEV (SAWy524). To confirm the effectiveness of JEN1tY180C in enhancing mevalonate import, it should also be tested independently of the optoMEV system already at this stage. Also is this performed in a background with the native Jen1pt? Ideally, JEN1tY180C ability to rescue growth should be tested in a clean $\Delta hmg1/\Delta hmg2$ without native JEN1. Again, this would also clarify if mevalonate can be imported/synthesised by other means.

As explained in our very first response to the reviewer, the experiment testing JEN1t(Y180C) independently of OptoMEV, requested by the reviewer, is in new Figure EV2E, which shows that JEN1t(Y180C) confers mevalonate-dependent growth to a clean $\Delta hmg1/\Delta hmg2$ strain, beyond the OptoMEV background. Also, as previously explained, we argue that the second experiment requested by the reviewer: "Ideally... without native JEN1" is unnecessary. The native *JEN1* is well-known to be transcriptionally (doi:10.1128/AEM.70.1.8-17.2004) and post-translationally (doi: 10.1074/jbc.M109.008318) repressed in glucose media so it is not expressed in our experiments. Furthermore, we show in separate experiments that expression of a wildtype version of the glucose-insensitive JEN1t without the Y180C mutation is unable to support growth of the double $\Delta hmg1/\Delta hmg2$ OptoMEV strain even when supplemented with mevalonate

(Figures 3B and EV2D). Therefore, this set of experiments strongly supports that mevalonate importation occurs through the evolved *JEN1t(Y180C)* and not *JEN1t* or the endogenous *JEN1*.

1.260 Maybe the author can clarify the advantages of using a saturation mutagenesis library (267 mutants) for position 180 given that testing 19 other strains (each with a different amino acid at this position) would be sufficient already.

As the reviewer implies, we could have prepared 19 separate strains and tested them individually. However, it was much faster, more economical, and less laborious to prepare a saturation mutagenesis library and screen it with at least 10-fold coverage, exploiting the enhanced growth phenotype we were screening for. This approach also helped demonstrate the versatility of the OptoMEV strain as a platform to screen mutant libraries for enhanced growth. This clarification has been added to the manuscript (**Lines 264-267**).

1.277 Fig. 3B. This data is discrete (different MVA concentration, light/dark) and should be represented through bar plots and not as connecting lines. Fig. 3d Legend for losanges and circles should be displayed on the graph for better readability.

We agree. The data in Figure 3B is now represented with bar plots.

The legend for losanges and circles is now displayed on the graph of Figure 3D.

1.380 Please add missing significance comparisons when relevant and indicate as n.s. when not statistically significant. In Fig. 4D, can the author also clarify what is $\delta::aaFS$ and why is it shown here?

The statistical significance of data comparisons has been added to Figures 2D, 2E, and 3B. For the sake of visual clarity and the journal's requirement to show exact p-values, we only show key statistical measurements. We regret the confusion about $\delta::aaFS$. This refers to the integration of the farnesene synthase aaFS in delta-integration sites of the yeast genome. This explanation has been added to the legend of Figure 4 of the revised manuscript (**Line 1135-1137**).

1.380 Fig. 4 should be complemented with additional data regarding the fermentation process. Fermentation time, starting OD, growth curve etc. It is difficult to judge from these bar plots the actual contribution of *JEN1t*. Panel B/D: I am confused by the strain genotype in this figure. Is the native *JEN1t* present there? If this is the case, then what's the contribution of the native vs mutated *Jen1pt* in farnesene increase?

Additional details of the fermentation experiments in Figure 4 have been added to the figure legend (**Lines 1128-1131**). These experiments involved high cell density fermentations for 48 hours (as described in the methods and now the figure legend), so they have no growth curves associated with them. The bar plots in figures 4B, C, and E show farnesene production in strains expressing different *JEN1t* variants when fed increasing concentrations of mevalonate. The more active variant *JEN1t(Y180G)* is more sensitive to lower concentrations of mevalonate. The

control experiment using a strain containing only *JEN1* (SAWy736; Appendix Figure S4) shows undetectable improvement in farnesene production when fed mevalonate. To help readers identify the different genotypes in this figure, we have added their basic description as a figure legend on each plot.

1.431 The microbial consortium presented here goes against the cooperation and complementation principles associated with microbial consortia such as division of labour, where the production pathway is typically split between members. Here, both strains are able to produce the full farnesene pathway which limits the relevance of using such a consortium. This translates into a moderate (yet significant) increase in farnesene production. I am not convinced that this increase in farnesene titers is due to mevalonate exchange for two reasons: 1. In the absence of growth curves, this increase could be related to the nature of the growth competition between different ratio of upstream/downstream strains. 2. Mevalonate exchange between members of the consortia is inferred but not proven. By which mechanisms would mevalonate be exported from the upstream strain. It is represented by a simple arrow in Fig. 5B without further information. Is it through the native *JEN1*?

It could be argued, perhaps, that our consortia do not clearly exploit the most common justification for consortia; that is division of labor. However, several other justifications for using consortia have been accepted, including to avoid toxic intermediates, reduce metabolic crosstalk, or to allow modular pathway optimization. In this case the co-culture is used to reassimilate a wasted secreted precursor that cannot be imported in a monoculture due to uphill gradients. This is supported by the fact that our co-cultures produce significantly more farnesene than either monoculture (Figure 5C). We now add this explanation in the revised manuscript (**Lines 405-408**).

The first concern of the reviewer, regarding growth competition between the two strains, can be ruled out by the way we executed our experiments. To avoid the type of confounding effects based on growth differences or competition, all our experiments were conducted in high cell density fermentations, in which the two strains were pre-grown separately and then combined in precisely measured ratios. This is explained in the Methods section with additional details (**Lines 739-752**). We also confirmed that these ratios were maintained throughout the fermentations with minimal drift, as shown in Figure EV4. To avoid this confusion, however, we have added new text to make this point clearer (**Lines 423-427**).

The second concern of the reviewer, regarding the mevalonate exchange between consortia members, can be addressed in parts. As we argue in our response to their second overall remark above, it is common practice and widely accepted in the field to use genetic evidence to demonstrate the role of a transporter in the assimilation of a metabolite (see for example: <https://doi.org/10.1186/s13068-014-0168-9>, <https://doi.org/10.1186/s12934-020-01354-9>, <https://doi.org/10.1073/pnas.1311970111>, <https://doi.org/10.1073/pnas.1010456108>, or <https://doi.org/10.1021/acssynbio.1c00229>). This is in line with the evidence we provide in this study for the ability of *JEN1* mutants to import mevalonate. Regarding mevalonate exportation, it has long been observed by us and others that yeast can naturally secrete mevalonate when

produced in excess (see for example: <https://doi.org/10.1093/jimb/kuab050> or <https://doi.org/10.1186/s12934-016-0447-1>). In this study, we were able to measure significant mevalonate secretion from the upstream strain, but not the downstream strain, of the consortium (Figure 5D). In a recent study, we found that mevalonate export in yeast is mediated by more than one endogenous transporter, and is not affected by *JEN1* disruption (<https://doi.org/10.1016/j.biotno.2024.10.001>). Therefore, while we do not fully understand the mechanism(s) of mevalonate secretion in yeast, it is undisputed that it occurs. Also undisputed is the fact that yeast is unable to significantly import mevalonate as demonstrated in this study. In summary, previous studies have established the ability of yeast to secrete mevalonate, while in this current study we present evidence that mevalonate can also be imported, and therefore exchanged between strains, but only when a *Jen1pt* mutant is expressed in the importing strain. To avoid this confusion, however, we have added new text to the manuscript explaining the points made above (**Line 397-402**).

Minor points:

1.186 "we introduced a glucose-insensitive mutant of *JEN1*". The authors could clarify the reasoning for selecting a glucose-insensitive mutant of *JEN1*. I.e. Presence of glucose leads to *Jen1* endocytosis.

We have added the following clarifying text: "However, because *Jen1p* is naturally repressed by glucose-mediated endocytosis and degradation, and glucose is such a preferred substrate in many applications, we introduced a glucose-insensitive mutant of *JEN1*, called *JEN1t*" (**Lines 183-185**).

Supp 1.52 / Fig. S3. Please change p1-p3 passage naming to something else to avoid confusion with p1 and p2 plasmids from OrthoRep. Also, meaning of "MVA" is missing. I suppose this represents 10mM mevalonate?

Thank you for the suggestion. We have renamed p1-p3 to "passage 1 – passage 3" throughout the manuscript. Also, in Appendix Figures S2 (formerly called Fig. S3), we now specify that "MVA" refers to growth conditions supplemented with 10 mM mevalonate.

Supp Table S6. Shouldn't yEZ118 be SAWy118 in the strain description of SAWy119. Double check typos in strain description.

We thank you very much for catching this mistake. This has been corrected, and we have doublechecked the accuracy of our tables.

1.314/316s typos: This should read *JEN1t*. Make sure this is corrected throughout the text

Thanks again for so carefully reading our manuscript. This typo is now corrected.

1.455 Fig. 5A and 5B Nomenclature is not consistent between subpanels. Is it upstream mevalonate pathway or upper mevalonate pathway?

The mevalonate pathway is subdivided into “upper” and “lower” mevalonate sub-pathways. The consortia, on the other hand, are divided into “upstream” and “downstream” strains. To avoid confusion, we will no longer use the terms “Upstream mevalonate pathway” or “downstream mevalonate pathway” in Figure 5 or the rest of the manuscript. .

Reviewer #2:

This is overall a strong study that adds both a general new tool/approach for continuous evolution (i.e., optoRep with optogenetic control of selection strength) and makes a notable engineering discovery of a new transporter for mevalonate based on JEN1. The work is very well written with clear logic. The work is also complete both in characterization of evolved transporters and their application in farnesene production, especially the microbial consortia format which is a smart and well-motivated variation. Experiments are complete and appear rigorous. Overall, I believe the work described and level of analysis carried out would be interesting to and enjoyed by MSB readers.

We thank the reviewer for their supportive comments

I have some minor issues to raise to guide revision...

1) I am not sure "gain-of-function" is the best term for what the authors mean. The authors are highlighting a case where a function is gained in a gene that did not have that function before (i.e., going from "zero to something"). But I believe gain-of-function is also commonly used to include cases where function is enhance rather than emerging from none (i.e., going from "something to something more"). The authors rightfully wish to emphasize the fact that in their experiment, JEN1 had no detectable activity for mevalonate before but found a mutation that now gives it activity, which is indeed a unique, interesting, and noteworthy outcome. The fact that JEN1 has no activity also motivates the need for an ontogenetic control strategy to titrate down the function of another gene to drive selection. Perhaps the better term is neofunctionalization? I'm sure there is some vagueness in how gain-of-function is used, given it is also part of colloquial language. At the very least, I recommend the authors consider this point. If they agree that gain-of-function is not the optimal term, the authors should look at the few instances in which it is used in their manuscript and modify accordingly.

We thank the reviewer for the suggestion. After careful consideration, we decided to adopt the term “neofunctionalization” throughout the manuscript. However, we do not use it instead of the term “gain-of-function”, which is generally more familiar to most readers, but rather use it as a qualifier to disambiguate the outcome of our gain-of-function mutations. For example, we now say: “OptoRep was able to select for gain-of-function mutations that neofunctionalize *JEN1* as an effective mevalonate importer” (**Lines 131-133**).

2) It would be helpful to explain why the MCT1 point mutant transporter that works in mammalian cells was not used here as the starting point for evolution, given it has some low activity in yeast. Also, it is interesting that we are also looking at a single aromatic to cysteine mutation for MCT1. Is that a coincidence or is it a structurally homologous position?

Indeed, we considered using MCT1 to evolve a *de novo* mevalonate importer in yeast. However, we found that the efficiency with which this importer is trafficked to the yeast plasma membrane is extremely low, explaining why its activity in yeast is also very low. Although evolving MCT1 with orthoREP should in principle help improve its membrane localization, this would require overcoming two major hurdles: *de novo* (or at least improved) mevalonate importation; and improved protein trafficking. Understanding this double challenge was crucial in our decision to focus on *JEN1*, which at least has already evolved to be efficiently trafficked to the membrane. We have included this explanation for why we chose to work on *JEN1* over MCT1 in the Introduction section (**Lines 117-122**) – thank you for the suggestion.

We do not believe the mutation from aromatic residue to cysteine in both *JEN1*t and MCT1 is a coincidence. Indeed, as we discuss in the Discussion section (**Lines 512-528**), we believe the aromatic residue in both proteins plays a similar role of orienting the substrate to interact with a conserved arginine. Therefore, mutation to a smaller residue (cysteine) increases the size of substrate channel, allowing mevalonate to be transported. We also do not believe the specific mutation to cysteine in both proteins is a coincidence, but the consequence of having to make a minor one-nucleotide substitution in the DNA sequence from A to C (for Phe; [https://www.jbc.org/article/S0021-9258\(18\)50064-8/pdf](https://www.jbc.org/article/S0021-9258(18)50064-8/pdf)) or from A to G (for Tyr).

3) I understand why HMG2 was deleted. But I believe it should be made more clear both textually and in Figure 1. On first glance, I was confused why Figure 1D showed good growth in the dark for the *deltahmg2* strain. After some thinking, I figured out that the *deltahmg2* growth curves were simply the more matched control for optoMEV strains since those need to have *deltahmg2* deleted as only *hmg1* was put under optocontrol but *hmg2* is sufficient to make mevalonate even when *hmg1* is not expressed. However, I think this can be made immediately clear to the reader by adding in the main text and figure legend. And by expanding the strain cartoons in 1A and B.

We agree that this could be confusing for readers and have added a description of the necessity for *HMG2* deletion on the main text (**Lines 146-147**) and the legend of Fig 1 (**Lines 1063-1064**) as suggested by the reviewer.

4) It would be good to have more discussion and information about the dependence of the [relationship between light exposure and selection strength] on [cell culture density].

We have more clearly explained how light pulsing is an effective way to tune the level of expression of essential genes, and thus cell growth (cell culture density), and the selection pressure to evolve proteins (**Lines 192-196**).

5) I thought it was smart that the authors mutated Y180 after finding Y180C from OrthoRep in order to sample other amino acids at that position that are not accessed due to the biased nature of OrthoRep's TP-DNAP1-4-2. This resulted in the discovery of the Y180G mutant that was ultimately superior. It might be helpful for the authors to note that OrthoRep systems with higher transversion rates are available (Rix et al., 2023, which the authors cite) to make the Y to C to G path more likely. It might also have been the case that decreasing mevalonate after going to dark in the evolution experiment would have found C to G even though it is a less common mutation due to TP-DNAP1-4-2's bias.

We thank the reviewer for the kind comment and the suggestion. We now mention that the new OrthoRep system with its reduced bias could have helped find the Y180G mutant (**Lines 501-503**). We also mention the possibility that the Y180G mutation might have been eventually found by further reducing mevalonate concentrations and increasing the number of passages, but emphasize that the main advantage of OptoRep is its application in scenarios in which the protein targeted for mutagenesis has undetectable levels of the desired activity (**Lines 506-511**).

7) I appreciate the docking experiments. But it should be noted that they are using an alphafold structure. The extent to which this creates liabilities remains to be worked out in the field. I would therefore recommend that the authors comment that readers should keep this in mind, even though what is docked seems sensible.

This caveat has been noted (**Lines 322-324**).

8) The authors should cite and give a brief note to this paper on transporter evolution when discussing the context of their work: <https://elifesciences.org/reviewed-preprints/93971>

We thank the reviewer for this great suggestion. We now draw parallels between this previous study (now cited in Reference 36) and our own (**Lines 487-492**).

9) One reality of the study is that in retrospect, the transporter single mutation could have likely been found by just using an error-prone PCR library on JEN1 and selecting for tolerance of HMG1/2 knockout in a population of JEN1 mutants plus mevalonate. Such an approach would have required neither OrthoRep nor optogenetic titration of the selection. Of course, this is only a realization in retrospect. But I think it would be helpful to mention this both as a nuance for readers to appreciate and a springboard to a further discussion of what types of adaptation or neofunctionalization would only be accessible with the optorep approach - ones requiring complex multi-mutational pathways or more complex selection histories involving fluctuating pressures, etc.

We have added a retrospective note and discussion to the revised manuscript (**Lines 477-483**).

Reviewer #3:

Summary

Wegner et al. apply a combined approach of the continuous directed evolution system OrthoRep, optogenetics (OptoRep) and classical directed mutation to evolve a mevalonate importer from a truncated version of the monocarboxylate transporter JEN1. No mevalonate importer has yet been identified in yeast and a mammalian mutant transporter has only low activity in yeast. By evolution of a mevalonate importer for yeast, the authors prove the power of OptoRep. The evolved transporter is then used to support isoprene production in yeast (consortia).

General remarks

The manuscript is well written and understandable. The authors outline the engineering strategy well which makes it easy to follow it. In addition, the data presentation is comprehensive and in a reasonable format. The main value of this manuscript might be the application of optogenetics in OrthoRep to finetune expression of metabolites which (yet) cannot be important into cells. This allows to evolve transport activities which have not been engineered yet. The authors demonstrate and present this concept very well. It adds a useful feature to the OrthoRep tool kit. From an industrial point of view, evolving a mevalonate transporter might be of limited usefulness, however, it is a good proof-of-concept target to demonstrate the power of the newly developed OptoRep approach.

We thank the reviewer for the supportive comments.

Major points

1. Using OrthoRep, the authors performed continuous directed evolution of JEN1 for three passages. This is a very short time to evolve a gene. As far as I can see, the evolved strain does not perform as well as the wildtype (around 70%), and it might be reasonable to perform the evolution for a couple more passages. Evolving the improved mutants (Y180G or Y180A) further in OrthoRep might have provided another strategy to improve the importer. Have the authors considered further improvement of the importer?

The reviewer is correct, we could have extended the number of passages to try to further improve the activity of the Y180C and therefore cell growth. However, it is quite remarkable that the same single point mutation emerged in three independent evolutionary campaigns to yield an impressive ~70% growth of the wild-type strain. Therefore, while we initially considered doing more passages, we reasoned that doing so would more likely lead to the accumulation of additional superfluous mutations (needing subsequent laborious deconvolution to elucidate their relevance), rather than mutations needed to explore the best substitutions at Y180 (not easily obtained by OptoRep because of its well-known biases). Therefore, we chose to follow the initial evolution of Y180C with the saturation mutagenesis at this position. Indeed, our strategy paid off when we found that Y180G (still a single amino acid substitution) brought the strain even closer to the wild-type growth rate (~80%). With this efficiency, we chose to test the evolved transporter in a metabolic engineering application, rather than going after any marginal activity improvement that would only delay publication of our results without changing the main conclusions of the study. To address this comment, however, we have added our reasoning above to the results (**Lines 210-215 and 218-221**).

To answer the reviewer's question, the activity of the Y180G mutant was sufficient to show a significant effect in the microbial community designed to produce farnesene, so we did not find it necessary to further improve its activity. We now make this specific point in the revised manuscript (**Lines 334-337**). However, if this activity becomes limiting in future applications, we will absolutely consider further evolving the JEN1t(Y180G).

2. Did the authors compare the activity of the evolved transporter with the mutant form of the mammalian transporter, e.g. in growth assays? Is the evolved transporter more or less active than existing one(s)?

The comparison suggested by the reviewer would be unfair to MCT1 and therefore misleading because the trafficking efficiency of MCT1 to the plasma membrane of yeast is extremely low, which leads to very low mevalonate importation activity. We now mention this in the revised manuscript (**Line 117-119**).

Nevertheless, we very recently published a separate study focused on MCT1, in which we measured its mevalonate importation activity (<https://doi.org/10.1371/journal.pone.0312492>) This study is now cited in Reference 17. The results revealed that indeed the activity of MCT1 is much lower than that of the evolved Jen1t variants. While we avoid a direct comparison in this study (out of fairness to MCT1), we now leverage this previous result as an indirect comparison (**Lines 119-122**).

Minor points

1. I am surprised that the yeast strains level off in growth at OD 1. Is the optoMEV strain restrained in growth due to the optogenetic engineering?

The OD600 units reported correspond to the unaltered measurements using the units obtained from the Tecan plate reader, which are 10-fold lower than those measured with a spectrophotometer with a path length of 1 cm. Therefore, the yeast cultures actually level off at ~10 OD600 units measured in a spectrophotometer. To avoid confusion, all figure legends containing OD measurements specify that the units correspond to Tecan measurements, and the 10X factor mentioned above is explained in the Methods (**Lines 679-681**).

2. Figure 1D shows that the optoMEV strain can grow somewhat in the dark (no MEV supplementation). Given a starting OD of 0.01 (?) it made four doublings. Is this a result of MEV carryover from the population from which it was inoculated or the TF EL222 leaky?

The weak cell growth in the dark without mevalonate is most likely due to slight leaky expression of *HMG1* from P_{C120}, which has been previously reported. This explanation has now been added (**Lines 171-172 and 1062-1063**). If there was an appreciable carryover of mevalonate we would anticipate an earlier increase in growth, which would then level off once this mevalonate is consumed, which is not observed.

27th Mar 2025

Manuscript Number: MSB-2024-12530R

Title: Orthogonal replication with optogenetic selection evolves yeast JEN1 into a mevalonate transporter

Dear Prof Avalos,

Thank you for the submission of your revised manuscript to Molecular Systems Biology. We have now received the enclosed reports from the referees that were asked to re-assess it. As you will see the reviewers are now globally supportive and I am pleased to inform you that we will be able to accept your manuscript pending the following final amendments and response to reviewers:

- 1) In the main manuscript file, please include keywords to max. 5.
- 2) Please rename "Conflict of Interest" to "Disclosure and competing interests statement". We updated our journal's competing interests policy in January 2022 and request authors to consider both actual and perceived competing interests. Please review the policy <https://www.embopress.org/competing-interests> and update your competing interests if necessary.
- 3) References: Please correct the reference citation in the reference list to be alphabetical (not numerical). Where there are more than 10 authors on a paper, only the first 10 should be listed, followed by "et al.". Please check "Author Guidelines" for more information.
<https://www.embopress.org/page/journal/17574684/authorguide#referencesformat>
- 4) In the Methods, please take care of the following:
 - In the Methods heading, please remove "[LOOK AT SCOTT'S REVISED METHODS]"
 - Please remove the Reagents and Tools Table in the Methods section of the manuscript and upload it as a separate file choosing the file type "Reagent Table".
- 5) Please place individual sections of the manuscript in the following order: Title page - Abstract & Keywords - Introduction - Results - Discussion - Methods - Data Availability - Acknowledgements - Disclosure and Competing Interests Statement - References - Figure Legends - Expanded View Figure Legends.
- 6) For the figures and figure legends, please take care of the following:
 - Please note that the legends for figure EV 1 is not provided in the sequential manner (legend for figure EV 1 C is provided before legend of figure EV1 D). This needs to be rectified.
 - Please note that information related to n is missing in the legends of figures 2D, E; 3A.
 - Please note that the error bars are not defined in the legends of figures 2D, E.
 - Please note that the measure of center for the error bars needs to be defined in the legends of figures 1C, D, E, F.
- 7) Appendix file: Please upload the Appendix as a single PDF (no separate image files are needed) and add page numbers to the Table of Contents. Additionally the title page should contain "Appendix for [+ ms title]". Finally the references should be alphabetical with 10 authors + et al. (same as reference formatting in the main manuscript).
- 8) Funding: Please ensure that all funding sources are entered into the manuscript. Currently the National Science Foundation Graduate Research Fellowship Program (GRFP) (DGE-2039656) is missing from the manuscript.
- 9) Synopsis:
 - Synopsis image: Please provide a graphic that summarises the main findings of the manuscript on a glance and upload it as a high-resolution jpeg file 550 pixels wide x (300-600) pixels high.
 - Synopsis text: Please provide a short standfirst (maximum of 300 characters, including space), limit the bullet points to max. 5 and upload it as a separate .doc file. Please write the bullet points to summarise the key NEW findings. They should be designed to be complementary to the abstract - i.e. not repeat the same text. We encourage inclusion of key acronyms and quantitative information (maximum of 30 words / bullet point). Please use the passive voice.
 - Please check your synopsis text and image before submission with your revised manuscript. Please be aware that in the proof stage minor corrections only are allowed (e.g., typos).
- 10) Source Data: Source Data should be organized as a single source data file (zipped) per figure for main figures (all EV and/or Appendix figure Source Data can be included in a single folder), with the panels clearly visible in the folder structure e.g. all the Source data files for figure 1 need to be saved in a single folder and this needs to be zipped and then uploaded as "SD figure 1.zip" file.
- 11) As part of the EMBO Publications transparent editorial process initiative (see our policy here: https://www.embopress.org/transparent-process#Review_Process), Molecular Systems Biology will publish online a Peer Review File (PRF) to accompany accepted manuscripts. This file will be published in conjunction with your paper and will include the anonymous referee reports, your point-by-point response and all pertinent correspondence relating to the manuscript. Let us know whether you agree with the publication of the PRF and as here, if you want to remove or not any figures from it prior to publication. Please note that the Authors checklist will be published at the end of the PRF.
- 12) After your paper is published, we will promote it on social media. If you have any handles or hashtags for Bluesky you would like included, please let us know.
- 13) Please provide a point-by-point letter INCLUDING my comments as well as the reviewer's reports and your detailed responses (as Word file).

I look forward to reading a new revised version of your manuscript as soon as possible.

Yours sincerely,

Poonam Bheda, PhD
Scientific Editor
Molecular Systems Biology

Reviewer #1:

The authors have now addressed all my concerns. With the additional controls and edits, the manuscript has improved in clarity and quality.

Minor comments:

Appendix Fig. S1 legend: (p1, p2, p3) should read (Passage1, Passage2, Passage3)

I. 674 DPNI should read Dpnl

Reviewer #2:

I am satisfied with the revision and recommend acceptance.

Reviewer #3:

I am still not convinced that running OrthoRep for only three passages is sufficient. The argument that it might cause accumulation of superfluous mutations is redundant because it can only be proven when running the campaign for longer. Still, I acknowledge that improving the transporter to 80% of the wildtype growth might be good enough for a proof-of-concept study. Retrospectively, directed evolution might have been more reasonable because with three passages it is faster to set up and run and might be less biased about what mutations are introduced. I see this as the great advantage of OrthoRep - it can be run for long with a hands-off procedure once it is set up and running.

All my other comments were met, and I recommend the manuscript for publication.

Dear Dr. Bheda,

We are delighted that you will be able to accept our manuscript for publication in Molecular Systems Biology, pending some final revisions. Below, we provide our point-by-point responses to the editorial and reviewers' comments (in blue). **Please note that the line numbers referenced below correspond to the revised manuscript with tracked changes.** We wish to thank you and the reviewers again for their time and contributions.

- 1) In the main manuscript file, please include keywords to max. 5

Keywords have been added following the abstract (**Line 46**).

- 2) Please rename "Conflict of Interest" to "Disclosure and competing interests statement". We updated our journal's competing interests policy in January 2022 and request authors to consider both actual and perceived competing interests. Please review the policy <https://www.embopress.org/competing-interests> and update your competing interests if necessary.

We have updated the name and content of the competing interest section (**Line 821-823**).

- 3) References: Please correct the reference citation in the reference list to be alphabetical (not numerical). Where there are more than 10 authors on a paper, only the first 10 should be listed, followed by "et al.". Please check "Author Guidelines" for more information.

<https://www.embopress.org/page/journal/17574684/authorguide#referencesformat>

Reference style has been updated.

- 4) In the Methods, please take care of the following: - In the Methods heading, please remove "[LOOK AT SCOTT'S REVISED METHODS]"

This text has been removed (**Line 560**).

- a. Please remove the Reagents and Tools Table in the Methods section of the manuscript and upload it as a separate file choosing the file type "Reagent Table".

The table has been moved to its own file.

5. Please place individual sections of the manuscript in the following order: Title page - Abstract & Keywords - Introduction - Results - Discussion - Methods - Data Availability - Acknowledgements - Disclosure and Competing Interests Statement - References - Figure Legends - Expanded View Figure Legends

The manuscript sections are in the order requested.

6. For the figures and figure legends, please take care of the following:
- Please note that the legends for figure EV 1 is not provided in the sequential manner (legend for figure EV 1 C is provided before legend of figure EV1 D). This needs to be rectified.

The EV1 figure legend has been corrected to adhere to a sequential manner (Lines 1194-1203).

- Please note that information related to n is missing in the legends of figures 2D, E; 3A.

The missing n has been added to the legend for figure 2 (Line 1107). The missing n for 3A has also been added (Lines 1114-1118).

- Please note that the error bars are not defined in the legends of figures 2D, E.

Error bars are now defined (Line 1108-1109).

- Please note that the measure of center for the error bars needs to be defined in the legends of figures 1C, D, E, F.

The measures represent the mean OD₆₀₀ (Lines 1087 – 1088).

7. Appendix file: Please upload the Appendix as a single PDF (no separate image files are needed) and add page numbers to the Table of Contents. Additionally the title page should contain "Appendix for [+ ms title]". Finally the references should be alphabetical with 10 authors + et al. (same as reference formatting in the main manuscript).

The Appendix has been updated with a table of contents.

8. Funding: Please ensure that all funding sources are entered into the manuscript. Currently the National Science Foundation Graduate Research Fellowship Program (GRFP) (DGE-2039656) is missing from the manuscript.

This funding source has been added to the manuscript (Lines 814-815).

9. Synopsis:

- Synopsis image: Please provide a graphic that summarises the main findings of the manuscript on a glance and upload it as a high-resolution jpeg file 550 pixels wide x (300-600) pixels high.

A synopsis image has been provided.

- Synopsis text: Please provide a short standfirst (maximum of 300 characters, including space), limit the bullet points to max. 5 and upload it as a separate

.doc file. Please write the bullet points to summarise the key NEW findings. They should be designed to be complementary to the abstract - i.e. not repeat the same text. We encourage inclusion of key acronyms and quantitative information (maximum of 30 words / bullet point). Please use the passive voice.

A standfirst has been provided.

- c. Please check your synopsis text and image before submission with your revised manuscript. Please be aware that in the proof stage minor corrections only are allowed (e.g., typos).

The synopsis has been checked for accuracy.

10. Source Data: Source Data should be organized as a single source data file (zipped) per figure for main figures (all EV and/or Appendix figure Source Data can be included in a single folder), with the panels clearly visible in the folder structure e.g. all the Source data files for figure 1 need to be saved in a single folder and this needs to be zipped and then uploaded as "SD figure 1.zip" file.

The source data files have been organized as specified.

11. As part of the EMBO Publications transparent editorial process initiative (see our policy here: https://www.embopress.org/transparent-process#Review_Process), Molecular Systems Biology will publish online a Peer Review File (PRF) to accompany accepted manuscripts. This file will be published in conjunction with your paper and will include the anonymous referee reports, your point-by-point response and all pertinent correspondence relating to the manuscript. Let us know whether you agree with the publication of the PRF and as here, if you want to remove or not any figures from it prior to publication. Please note that the Authors checklist will be published at the end of the PRF.

We consent to the PRF publication and do not wish to exclude any figures.

12. After your paper is published, we will promote it on social media. If you have any handles or hashtags for Bluesky you would like included, please let us know.

@javalospu.bsky.social

13. Please provide a point-by-point letter INCLUDING my comments as well as the reviewer's reports and your detailed responses (as Word file).

See below for the response to reviewer's reports.

Reviewer #1:

The authors have now addressed all my concerns. With the additional controls and edits, the manuscript has improved in clarity and quality.

We thank the reviewer for the supportive comments.

Minor comments:

Appendix Fig. S1 legend: (p1, p2, p3) should read (Passage1, Passage2, Passage3)

This has been corrected.

1. 674 DPNI should read DpnI

This has also been corrected in Main Text and Supplementary Table (**supplied as a separate document to the main text**).

Reviewer #2:

I am satisfied with the revision and recommend acceptance.

We thank the reviewer for the supportive comments.

Reviewer #3:

I am still not convinced that running OrthoRep for only three passages is sufficient. The argument that it might cause accumulation of superfluous mutations is redundant because it can only be proven when running the campaign for longer. Still, I acknowledge that improving the transporter to 80% of the wildtype growth might be good enough for a proof-of-concept study. Retrospectively, directed evolution might have been more reasonable because with three passages it is faster to set up and run and might be less biased about what mutations are introduced. I see this as the great advantage of OrthoRep - it can be run for long with a hands-off procedure once it is set up and running.

All my other comments were met, and I recommend the manuscript for publication.

We agree that extended evolutionary campaigns is an exciting avenue to pursue in future studies, especially if multiple mutations are necessary to observe a dramatically improved activity (contrary to what was seen here).

We thank the reviewer for the supportive comments.

2nd May 2025

Manuscript number: MSB-2024-12530RR

Title: Orthogonal replication with optogenetic selection evolves yeast JEN1 into a mevalonate transporter

Dear Prof Avalos,

Congratulations on an excellent manuscript, I am pleased to inform you that your manuscript has been accepted for publication in Molecular Systems Biology. Thank you for your comprehensive response to referee concerns. It has been a pleasure to work with you to get this to the acceptance stage.

Yours sincerely,

Sincerely,

Poonam Bheda, PhD
Scientific Editor
Molecular Systems Biology
